# *Onopordum acanthium* L. extract attenuates pancreatic β-Cells and cardiac inflammation in streptozocin-induced diabetic rats

Abdalmuhaimen Yusif Sharef[1], Bushra Ahmed Hamdi[1]*, Rafal Abdulrazaq Alrawi[1], Hiwa Omer Ahmad[2]

1 Department of Clinical Analysis, College of Pharmacy, Hawler Medical University, Erbil, Iraq, 2 Department of Pharmaceutical Chemistry, College of Pharmacy, Hawler Medical University, Erbil, Iraq

* bushra.hamdi@hmu.edu.krd

## Abstract

### Background and objective

Methanolic extract from *Onopordum acanthium* L. leaves (MEOAL) has been discovered to treat diabetic complications. The objective of this study is to evaluate the ameliorative role of MEOAL on pancreatic islet injury and myocardial inflammation in diabetic rats.

### Methods

Forty male Wister albino rats were allocated into five groups of eight rats each. Group A was the negative control group. Single intraperitoneal injection of streptozocin (50mg/kg) were used for the four experimental groups. Group B served as the positive control group. The rats in Groups C, D, and E received glibenclamide (5mg/kg), MEOAL (200, and 400 mg/kg) respectively, for eight weeks. Group C served as the standard drug group. High performance liquid chromatography (HPLC) and 2,2-diphenyl-1-picryl-hydrazyl-hydrate (DPPH) assays for antioxidant activity were conducted in MEOAL. In silico study, calculation of molecular binding energy (DG), and inhibition constant (pKi) of bioactive constituents in MEOAL were performed.

### Results

Administration of MEOAL significantly increases insulin content in β-cells with a marked enhancement of pancreatic islet structure, resulting in a significant reduction of blood glucose level and body weight loss. MEOAL treatment suppressed the increase of inflammatory cell score in myocardial tissue with an elevation of M2 –like macrophage. The phytochemical studies recorded the presence of six polyphenols, including catechin, kaempferol, syringic acid, p-coumaric acid, epicatechin and gallic acid in MEOAL. Moreover, the antioxidant activity of the extract was greater than that of standard ascorbic acid. The docking studies of the ligands Catechin, kaempferol and epicatechin with proteins showed high affinities with various targets related in β-Cells and cardiac inflammation.

**Data Availability Statement:** Data are available from the following link https://doi.org/10.6084/m9.figshare.21834111.v1.

**Funding:** The author(s) received no specific funding for this work.

**Competing interests:** The authors have declared that no competing interests exist.

## Conclusions

The attenuation of pancreatic β-Cells damage and cardiac inflammation by MEOAL could be attributed to the presence of Catechin, kaempferol and epicatechin which have high affinities with the receptors namely pancreatic alpha-amylase, glucokinase, COX-2, and COX-1.

## Introduction

Diabetes Mellitus (DM) is a multiple metabolic disorder that leads to abnormal metabolism in carbohydrates, fats, and protein, causing hyperglycemia and hyperlipidemia [1]. Millions of people have been affected by the development of diabetic complications, which is still a major endocrine disease [2]. It generally promotes the risk of macro and microvascular disorders in many organs, such as the heart, kidney, retina, and brain [3]. However, it has been shown that persistent hyperglycemia can promote glucose auto-oxidative and protein glycosylation, increase polyol and hexosamine pathways and induce protein kinase activation that causes changes in the inflammatory mediator's grade. Transitional mechanisms may produce reactive oxygen species (ROS) in DM, which directly contribute to the elevation of oxidative stress in various tissues [4]. ROS induces oxidative stress that plays a pathological role in the development and progression of diabetic complications [5].

Strptozocin (STZ) is known to have toxic effects on pancreatic β-cells and may be used as a potential inducer of oxidative stress [6]. The recent study referred to the use of STZ as a diabetic inducer in experimental animals. Intraperitoneal injection of STZ leads to shrinking of the islet of Langerhans cell size with severe architectural disarray. Moreover, it was found that the development of diabetic complications was associated with an increased risk of morbidity and mortality from cardiovascular disease [7]. It is found in a myocardial biopsy from diabetic animals that many cardiac morphological abnormalities, including cardiomyocyte hypertrophy, increased quantities of matrix collagen and, perivascular fibrosis were recorded [8]. In addition to mitochondrial dysfunction, endoplasmic reticulum stress and endothelial dysfunction were also observed [9].

In recent years, scientific attention has been focused on the search for safe and effective medicinal plants that can prevent or delay the development of diabetic complications [10–12]. The bioactive constituents of green plants modulate carbohydrate and lipids metabolisms, decrease glycemia and insulin resistance and minimize oxidative stress of DM due to their antioxidants' activities [2].

The present investigation is concerned with pancreatopathy and cardiopathy as a progressive consequence of diabetic complications and the study of potential plant extracts that contain natural products as antidiabetic agents or biological compounds, such as antioxidants that have a protective effect against the pathogenesis of the such disease.

*Onopordum acanthium* L. (*O. acanthium*) or cotton thistle, is a species that belongs to the family of Asteraceae used in Europe and Asia as a traditional medicine for different types of cancer and nervousness [13]. This plant has been applied and documented by traditional uses in various plants of the world, including its use as an antibacterial, anti-inflammatory, antimalarial and hypertensive agent [14, 15] due to its components with antioxidant properties, such as tocopherol, flavonoids, and phenolic acid [16, 17]. To the best of our knowledge, *O. acanthium* leaves contain lignans, triterpenoids, sesquiterpene lactones, sterols, flavonoids, and phenylpropanoids [13]. Those active compounds may promote glucose metabolism and improve pancreatic and cardiac morphology.

Since no previous studies were reported for anti-diabetic remedy of *O. acanthium* leaves, in this study we have evaluated the anti-diabetic efficacy of MEOAL in experimental diabetic rats. The main objective of this work is to investigate whether MEOAL attenuates pancreatic β-cells damage and cardiac inflammation through histopathological and immunohistochemical analysis in pancreatic and cardiac sections of diabetic rats. The identification of bioactive compounds and the antioxidant activity of MEOAL were studied using the HPLC and DPPH assay, respectively. Furthermore, in silico study we have identified bioactive compounds by which MEOAL exerts its effects.

## Materials and methods

### Plant extraction

Fresh leaves of *O. acanthium* were collected in April 2020 from the Kurdistan region, Iraq. The plant was identified by professor Jawher Saeed from Salahaddin University, Erbil, Kurdistan, Iraq, voucher Number: 3115. The leaves were washed, dried, at room temperature, cut into small pieces and then crushed. The powder (100 g) was macerated with a mixture of methanol and water (80:20, v/v) for seven days and then filtered with the Whatman No.3 filter paper. The obtained filtrate was concentrated using a rotary evaporator (Buchi rotavapor R114, Switzerland) at 40˚C. The yield of dried concentrated extract was achieved at 13% (w/w) and then stored at 4˚C for further use.

### High-performance liquid chromatography study

**Preparation of stock and standard solutions.** The stock solutions of polyphenolic compounds were prepared by dissolving the standard in methanol HPLC grade [18] at a concentration of 1 mg/ml. The stock solutions were further diluted with methanol to prepare serial concentrations of 10, 20, 30, and 40 μg/mL for preparing a calibration curve concentration versus absorbance.

**HPLC instrumentation and separation conditions.** A chromatography study was performed for phenolic and flavonoid compounds of MEOAL by using HPLC-UV/Visible (SYKAM, Germany) with S 2100 Quaternary Gradient Pump. Briefly, 10 μL of the sample was injected using an HPLC system equipped with S 5200 autosampler and then separated on a C18-ODS (25 cm × 4.6 mm) column. An isocratic solvent system was used comprising of: (A) methanol; (B) water; (C) 5% formic acid (Scharlau Chemie SA, European Union) at the ratio (70:25:5). The total run time for the injection was 20 min with the UV detection wavelength of 280 nm and using a flow rate of 1.0 mL/min. The quantitative concentration of the phenolic acids and flavonoid contents of MEOAL were measured by matching retention time and area under the peak. External standards were used for the documentation and quantification of compounds at ambient temperature. The results are expressed as μg/g of dry weight.

### Determination of antioxidant activity assay

The stable 2, 2-diphenyl-1-picrylhydrazyl radical (DPPH) assay method was used to determine the ability of MEOAL to scavenge the DPPH free radical. The DPPH (Merck, Germany) was prepared by dissolving 0.04 g in 100 mL of methanol [17]. A stock solution of vitamin C (Sigma-Aldrich, Germany) (Positive control) and an ethanolic **MEOAL** was prepared by dissolving 0.5 gm in 100 mL methanol-distilled water. Then, serial dilutions of vitamin C and extract were prepared at concentrations of 30, 60, 120, 250 and 500 μg/mL. An amount of 500 μL of each prepared concentration was separately mixed with 5 mL of DPPH. The mixed solution was incubated for 30 min at room temperature, and the absorbance was measured

against the blank at 517 nm wavelength with a spectrophotometer (UV-Vis Shimadzu). The percent of the DPPH scavenging effect was calculated by using the following equation:

DPPH scavenging effect(%) or inhibition(%)

$$= \frac{Absorbance\ (blank\ solution) - Absorbance\ of\ extract\ solution}{absorbance\ (blank\ solution)} \times 100 \qquad (1)$$

The concentrate of the extract (μg/mL) that inhibits 50% of DPPH ($IC_{50}$) represents the efficacy of the extract that scavenged or cleared 50% of the DPPH radical. The $IC_{50}$ was determined from the log concentration—(%) inhibition effect using a regression equation from the best-fit correlation line.

**Ethical approval.** All experimental protocols were approved by the Animal Ethical Committee from Hawler Medical University, Erbil, Iraq (approval number: 210720–111).

## *In vivo* experimental study

**Chemicals and reagents.** STZ was purchased from Glentham Life Sciences Ltd. (Corsham, UK). The tablet glibenclamide Sanofi (Daonil, France) was purchased from a local pharmacy. For the immunohistochemical study, insulin and MRC1 antibodies (CD206) were purchased from MyBioSource Inc. (San Diego, CA, USA). Chemical substances, buffers and diluents used for the immunohistochemical procedures were obtained from Dako, Denmark. All other chemicals used in the present study were of analytic grade and procured from Sigma-Aldrich, Germany.

**Experimental animals.** This study was carried out in the College of Pharmacy, Hawler Medical University in Erbil, Iraq, from September 2020 to June 2021. Forty male Wister rats (*Rattus norvegicus domestica*) 8–10 weeks old and weighing between 200g and 250g were used in this study. All were obtained from the animal house facility of Zakho University, Kurdistan Region, Iraq. The animals were maintained at a constant temperature (23 ± 1˚C) on a 12h dark/light cycle with free access to food and water.

**Acute toxicity study.** An acute oral toxicity test was conducted according to the Economic Cooperation and Development guidelines [19]. Twenty male Wistar rats were allocated into four groups (n = 5 per group). Groups 1–4 received a single oral dose of a MEOAL of 500, 1000, 1500, and 2000 mg/kg body weight, respectively. Mortality and general behavior were observed and recorded for 30 minutes, and periodically after 4 hours, 24 hours and, two weeks.

**Induction of DM.** Rats that were deprived of food pellets overnight were each treated with a single intraperitoneal injection of STZ at a dose of 50 mg/kg [20]. STZ was freshly prepared by dissolving it in citrate buffer 0.1 M, pH 4.5 at 4˚C. Then, the rats were supplemented with an oral 5% dextrose solution for 24 hours to prevent hypoglycaemic shock. On the 7th day after injection, fasting glucose levels were measured and a blood glucose level of ≥350 mg/dL was considered to be STZ- induced diabetic rats.

**Experimental design.** This study used 40 rats (8 normal control rats and 32 diabetic rats). After STZ-induced DM was established, the rats were allocated into five groups of eight rats each. For group A, a single dose of citrate buffer was injected intraperitoneally instead of STZ solution which was used for other groups (B, C, D, and E).

Animals began treatment after three days of diabetic confirmation.

Group A: Nondiabetic rats received 0.5 mL of normal saline and served the normal control group (NC) which was considered a negative control.

Group B: Diabetic rats received 0.5mL of normal saline and served a diabetic control group (DC) which was considered a positive Control.

Group C: Diabetic rats received a 5mg/kg body weight of a glibenclamide aqueous solution. This group served a reference (GLB group) [21, 22].

Group D and E: Diabetic rats received 200 and 400 mg/kg body weight of extract (200-MEOAL group) and (400-MEOAL group), respectively [17].

In all groups, the oral route of administration was achieved by using a specific rat feeding tube (gavage) daily for up to eight weeks.

**Determination of blood glucose and body weight.** Fasting blood glucose levels and body weight in all groups were determined throughout the experimental period at zero, two, four, and eight weeks of treatment. Body weight was recorded using Sartorius balance (Germany). Fresh blood samples were collected after 12 hours of fasting from the vein of the rat tail, and glucose level was determined by using glucose test strips with its glucometer (Accu-CHEK ® Active, Roche Diagnostics. Mannheim, Germany).

**Morphometric analysis.** At the end of the experiments, the animals were anaesthetized and killed by decapitation. The animals were immediately dissected, and the pancreas and heart were excised immediately and immersed in 10% formaldehyde solution (a process of tissue fixation) and later processed for histological and immunohistochemical studies.

**Histopathological study.** Fixed tissues from the pancreas and heart were processed using an automatic tissue processor. The tissues were dehydrated in ascending alcohol (80–100), cleared in xylene, and embedded in paraffin. Then paraffin blocks were sliced into 4μm serial cross-sections using a semi-automated microtome (Thermo Scientific). Following deparaffinization and rehydration, the sections were stained with hematoxylin-eosin [23]. Stained slides were photographed using the CCD digital camera (Olympus DP-12), which was attached to the light microscope (Olympus CX41) at a magnification of 40X. Histological changes in the pancreatic sections of all groups were observed and assessed by pathologists. While heart sections were scored for inflammation into four categories as follows [24]:

Score 0 = Normal cells with no inflammatory infiltrate, score 1 = 1–5 mononuclear cell infiltrates (minimal grade); score 2 = 6–20 mononuclear cell (mild grade); score 3 = more than 20 mononuclear cell (moderate grade); score 4 = multifocal coalescent inflammatory infiltrates (severe grade).

**Immunohistochemical analysis.** The details of immunohistochemical analysis for pancreas and heart tissues were summarised in Table 1. Sections were deparaffinized, rehydrated in alcohol, and then transferred into a diluted target-retrieved solution in a PT Link (Dako North American Inc.) for one hour. Next, sections were incubated with their specific primary antibodies for one hour at room temperature. After washing the samples, they were incubated with the corresponding HRP-labelled secondary antibody for 20 minutes. Samples were then washed with buffer and peroxidase activity was visualized by treating the slide with Di-Amino Benzidine plus chromogen for six minutes. Finally, the slides were counterstained with hematoxylin and cover-slipped for examination using a light microscope. Images of the immunostaining sections were analyzed by the Image J software (version 1.47v, National Institutes of Health, Bethesda, MD, USA) by using an intensity threshold that matched to the visually identified staining areas as closely as possible. The percentage area (%) of brown color was measured and averaged for each group [25]. All of these data were subjected to statistical analysis. Insulin and MRC1 antibodies were used to determine the percentage area of insulin in the

**Table 1. Characteristics of immunohistochemistry staining study.**

| Organ | Primary antibody | Specificity | Dilution |
|---|---|---|---|
| Pancreas | Proinsulin Antibody | Rat, Murine | 1:50 |
| Heart | MRC1 Antibody | Rat, mouse, human | 1:50 |

pancreas and macrophage 2 (M2) in the heart section, respectively. Data were expressed as the percentage area of insulin and MRC1 (CD206) positive cells in the pancreas and myocardial tissue, respectively (Table 1).

**In silico molecular docking studies.** The two-dimensional(2-D) structures of the ligand molecules (A-G) (Fig 1) were built using chemdraw professional 16.0 and converted to 3-dimensional (3-D) structures using Chem3D 16.0 module and saved as a pdb format structures (http://www.cambridgesoft.com/). The ligand was optimized by adding Geister

**Fig 1. The D2- structures of A: *Catechin* B: *Epicatechin* C: *Gallic Acid* D: *Glibenclamide* E: *kaempferol* F: *P-Coumaric Acid and G: Syringic acid* docked studied compounds.** https://doi.org/10.6084/m9.figshare.21828843.v1.

charges and hydrogen and the pdbqt format of the ligands were prepared with AutoDock Tools 1.5.7 [26].

The ligand molecules were then used as input for AutoDock Vina (https://vina.scripps.edu/) to carry out the docking simulation.

Protein molecules of Human pancreatic alpha-amylase (PDB ID: 2QV4) [27], Human glucokinase (PDB ID: 1V4S) [28], COX-2 (PDB ID: 1CVU), and COX-1 (PDB ID: 3N8Y) [29] were retrieved from the protein data bank (http://www.rcsb.org/pdb/) [30]. The grid dimensions were set at 17.4 x 61.8 x 15.9 (PDB ID: 2QV4), 27.5 x 1.5 x 68.6 (PDB ID: 1V4S), 28.1 x 29.1 x 40.7 (PDB ID: 1CVU), and 39.9 x -52.0 x -1.6 (PDB ID: 3N8Y) according to the coordinates x, y, and z, for the target binding sites. The water molecules were removed from the receptors and polar hydrogen and Kollman charges were added. The pdbqt format of the receptors were prepared by AutoDock Tools 1.5.7. AutoDock Vina was compiled and run under Windows 10.0 Professional operating system. Discovery Studio 2021 was used to deduce the pictorial representation of the interaction between the ligands and the target protein.

There is a traditionally calculation of the basic equations of enzyme kinetics of the Lineweaver–Burk assay extrapolated on 2D for enzyme-inhibitor complex's inhibition constant (Ki).

Sophisticated arithmetic and analytical in silico algorithms have been proposed to compute the inhibition constant (Ki) parameter [31].

**Statistical analysis.** All the values were expressed as mean ± standard error of the mean (SEM). The data were analyzed using a one-way analysis of variance (ANOVA) followed by post hoc Tuckey test. All analyses were performed using SPSS version 22. The p-value of $\leq 0.05$ was considered statistically significant (IBM SPSS Statistics for Windows, version 22 (IBM Corp., Armonk, N.Y., USA).

## Results

### HPLC study of the extract

Phytochemical analyses of MEOAL revealed the presence of six phenolic compounds with various quantities namely; catechin, kaempferol, syringic acid, p-coumaric acid, epicatechin, and gallic acid. The estimation of quantitative amounts shows the methanolic extract contains the highest concentration of gallic acid (98.4 μg/g of extract), while the syringic acid accounted for a lower concentration (20.6 μg/g of extract) (Table 2). (Fig 2) shows the chromatograms of plant extract and the peak area of each compound, and Table 2 shows the net results of the data of HPLC.

### Antioxidant activity

The antioxidant activity of MEOAL was determined by assessing the scavenging of the DPPH radicals. Table 3 shows that the $IC_{50}$ of the extract is 96.7 μg compared with the $IC_{50}$ of ascorbic

**Table 2. Quantitative analysis for MEOAL by HPLC.**

| Phenols | λ (nm) | Retention time (min) | Concentration (μg/g) | Regression equation | Squared correlation coefficient ($R^2$) |
|---|---|---|---|---|---|
| Catechin | 280 | 2.500 | 50.69 | y = 51.31x | 0.9997 |
| Kaempferol | 280 | 3.100 | 89.7 | y = 12.1x | 0.9994 |
| Syringic acid | 280 | 5.200 | 20.6 | y = 12.3x | 0.99964 |
| P-Coumaric acid | 280 | 6.500 | 77.4 | y = 32.6x | 0.9999 |
| Epicatechin | 280 | 7.300 | 26.8 | y = 5.6x | 0.9999 |
| Gallic acid | 280 | 8.100 | 98.4 | y = 32.1x | 0.9994 |

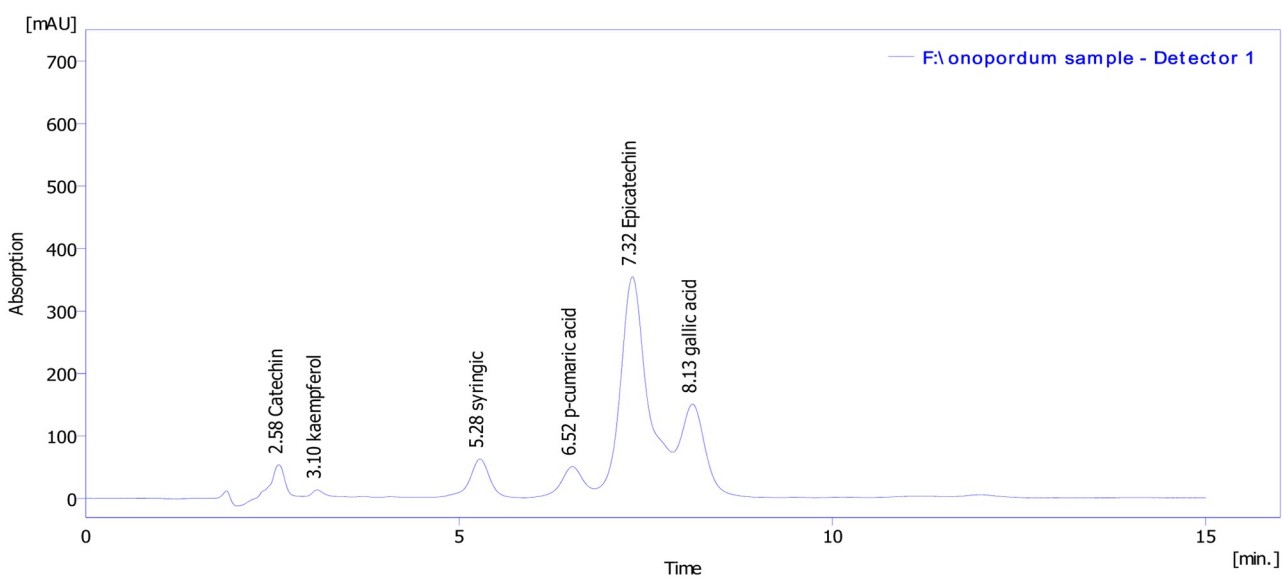

**Fig 2. Chromatographic profiles of plant extract.** https://doi.org/10.6084/m9.figshare.21828834.v1.

acid of 215.5 μg. This observation indicates that the OAE provides 2.2 times more antioxidant effects than ascorbic acid (Table 3).

## *In vivo* assessment of acute toxicity testing

The MEOAL showed no toxic reactions or lethality during two weeks of observation. The extract at the doses of (500, 1000, 1500, and 2000 mg/kg) did not produce a significant behavioral change. Therefore, a subsequent biological evaluation was carried out using the methanol extract from the leaves of the plant at concentrations of 200 and 400 mg/kg body weight.

## *In vivo* experimental studies

**1. MEOAL attenuates body weight loss.** Diabetic rats from the current study presented many clinical disorders, such as weight loss. As illustrated in Table 4, there was a significant decrease in the weights of diabetic control rats compared to normal control rats after two, four, and eight weeks. It indicated that the diabetic rats that were administrated with glibenclamide or extract at both doses did not exhibit significant weight gain compared to the normal control group. However, the same groups showed a significant increase (P<0.05) in weight gain over the second, fourth and eighth weeks of treatment compared to the diabetic control group. Interestingly, the weight gain improvement in 400-MEOAL group was higher than in other diabetic-treated groups. This indicates that 400mg/kg of extract treatment was able to normalize the body weight of diabetic rats at the end of eight weeks of administration (Table 4).

**Table 3. Antioxidant activity of MEOAL.**

| Sample | DPPH scavenging activity IC50 |
| --- | --- |
| MEOAL | 96.7 |
| **Ascorbic acid** | 215.5 |

**Table 4. Effects of GLB or MEOAL on the body weight gain of STZ-induced diabetic rats at different interval periods in different groups.**

| Groups | 2-weeks | 4-weeks | 8-weeks |
|---|---|---|---|
| **Normal control (NC)** | 40±0.597[d] | 59.62±1.164[c] | 129.375±0.88[d] |
| **Diabetic control (DC)** | -3.25±0.59[a] | -9.62±1.926[a] | -40.125±3.01[a] |
| **Glibenclamide (GLB)** | 7.87±1.64[c] | 13.00±2.604[b] | 32.00±3.620[b] |
| **200- MEOAL** | 4.124±0.398[b] | 16.87±2.495[b] | 36.875±8.8[b,c] |
| **400- MEOAL** | 4.12±0.833[b] | 22.50±4.818[b] | 61.625±11.46[c] |

The results are expressed as mean ± SEM (n = 8). The p-value was calculated using ANOVA with Turkey's multiple comparisons post Hoc test. The same letters mean no significant differences, the different letters a, b, c, and d mean significant differences at ≤ 0.05.

**2. MEOAL attenuates blood glucose level.** Blood glucose levels were significantly lower in the glibenclamide or MEOAL -treated groups compared to the values of the diabetic control group after two, four and eight weeks of administration. Extract treatment at a dose of 400mg/kg achieved a more antidiabetic effect than 200 mg/kg or glibenclamide treatment. There was an abrupt decrease in blood glucose level during the eighth week (53.6%) of MEOAL -400mg/kg treatment compared to their first day values. Glibenclamide, or extract -200mglkg treatment, resulted in 34.8% and 31.7% reductions in blood glucose levels, respectively, over the eight weeks (Table 5).

**3. Histopathological examination.** *3.1. Histopathological examination of pancreatic tissue.* The photomicrographs of hematoxylin & eosin (H&E) staining of pancreatic tissue of normal control rats showed normal histological architecture of the pancreatic acini and islets of Langerhans with centrally located β-cells (. In contrast to the normal control group, diabetic control rats revealed abnormal disruption of the pancreatic islet of Langerhans that is characterized by necrosis, cellular degeneration, and scattered mixed inflammatory cells infiltration. An apparent reduction in the number of β-cells were seen (Fig 3B1 & 3B2). However, diabetic animals treated with glibenclamide or MEOAL 200 mg/kg showed moderate improvement in the previously described histopathological lesions of the pancreatic tissue (Fig 3C & 3D). The diabetic group that received MEOAL 400mg/kg showed marked attenuation in pancreatic tissue damage and enhancement with almost cellular regeneration, increasing the number of pancreatic islets with relatively normal architecture, and exhibiting a protective response to a large dose of extract (400 mg/kg).

*3.2. Histopathological examination of heart myocardial tissue.* Histopathological findings by (H&E) carried out for heart muscle was shown in (Fig 4). The normal control rats revealed normal myocardial architecture with no inflammatory cells (Fig 4A). On the contrary, the control diabetic animals showed disordered myocardial cells, fatty change, edema, and pyknosis of

**Table 5. Effects of GLB or MEOAL on the blood glucose of STZ-induced diabetic rats at different interval periods in different groups.**

| Groups | First day | 2-weeks | 4-weeks | 8-weeks |
|---|---|---|---|---|
| **Normal control (NC)** | 79.25±1.04[a] | 78.5±0.62 [a] | 79.37±1.30 [a] | 80.12±1.57 [a] |
| **Diabetic control (DC)** | 405.12±0.93[b c] | 410.75±1.0 [c] | 421.12±1.49 [d] | 450.62±2.16 [d] |
| **Glibenclamide (GLB)** | 408.5±1.26 [c d] | 383.62±3.18 [b] | 368.25±3.83 [c] | 266.78±11.65 [c] |
| **200- MEOAL** | 403.12±1.23 [b] | 371.75±4.42 [b] | 331.25±6.59 [b] | 275.25±11.17 [c] |
| **400- MEOAL** | 410.12±0.87 [d] | 369.0±7.29 [b] | 312.37±11.78 [b] | 190±18.0 [b] |

The results are expressed as mean ± SEM (n = 8). The p-value was calculated using ANOVA with Turkey's multiple comparisons post Hoc test. The same letters mean no significant differences, the different letters a, b, c, and d mean significant differences at ≤ 0.05.

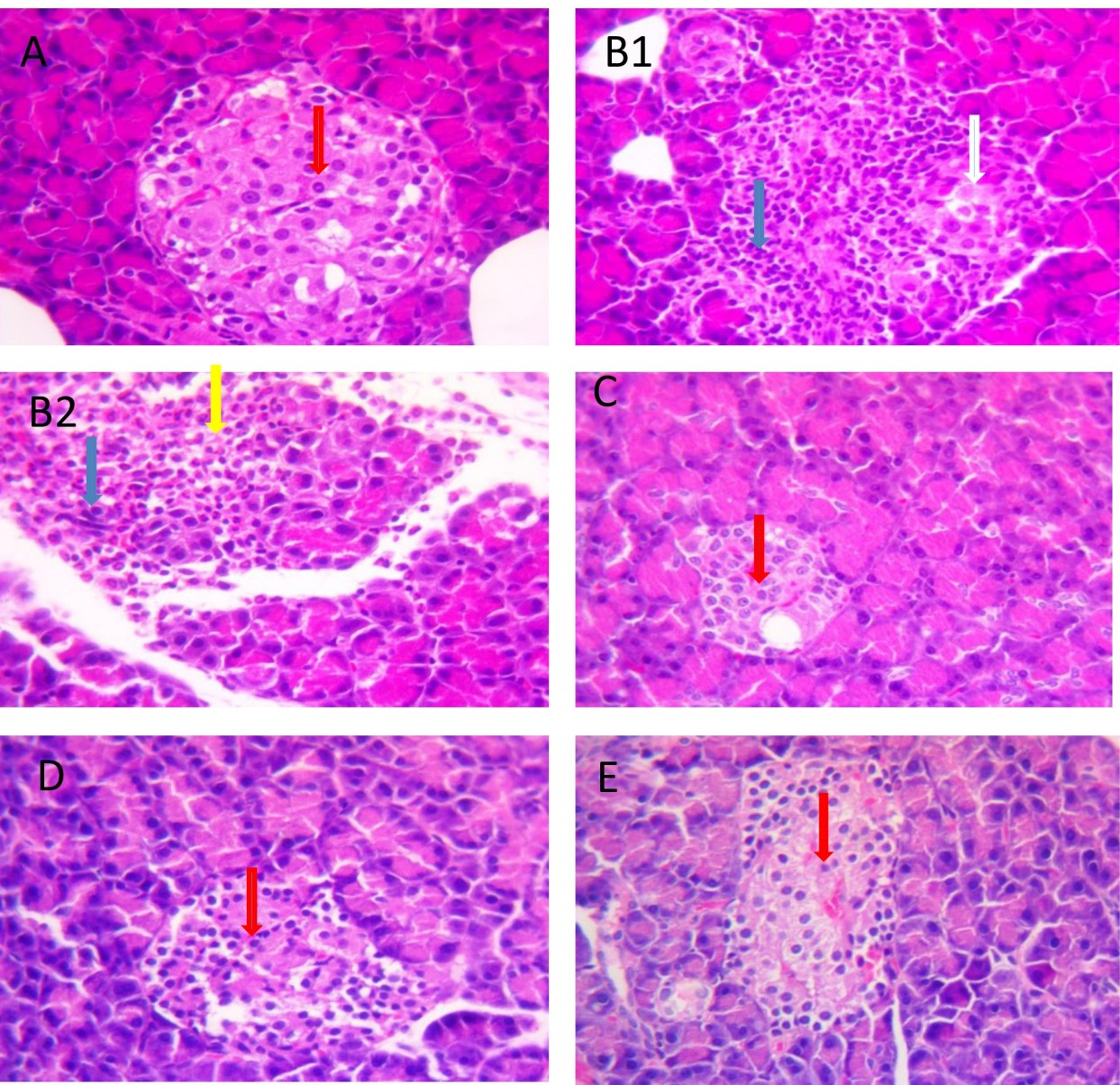

**Fig 3. Effect of GLB or MEOAL on histopathological changes of pancreatic tissue in different groups.** Hematoxylin and Eosin staining sections under 40X magnification. https://doi.org/10.6084/m9.figshare.21828810.v2. Photomicrograph of the pancreas from; (A) Normal control group showing the normal histological architecture of the pancreas with intact and centrally located β-cells (red arrow); (B1&B2) Diabetic control group showing distortion of the islet with prominent necrotic β-cells (yellow arrow), Degenerative alterations in β-cells (white arrow) and inflammatory cells infiltration mainly lymphocytes (blue arrow) have been observed; (C) Diabetic GLB (5mg/kg) treated group showing partial recovery of the tissue with a moderate increase in the number of the β-cells (red arrow); (D) Diabetic MEOAL (200mg/kg) treated group showing partial recovery of the tissue with a moderate increase in the number of the β-cells (red arrow); (E) Diabetic MEOAL (400mg/kg) treated group showing a dramatic recovery of the tissue with a marked increase in the number of the centrally located β-cells (red arrow). Abbreviation: MEOAL; *O. acanthium* extract. GLB: glibenclamide.

myocytes. In addition, cardiac fibrosis and hypertrophy of myocytes were observed (Fig 4B1 & 4B2). There was a significant increase ($p < 0.05$) in the score of inflammatory infiltration cells compared to the normal control group (Fig 4B1, 4B2 & 4F). On the other hand treatment with MEOAL (200 or 400 mg/kg) ameliorated most myocardial abnormalities (Fig 4D & 4E). However, some abnormal alterations were still seen in myocardial of diabetic rats treated with

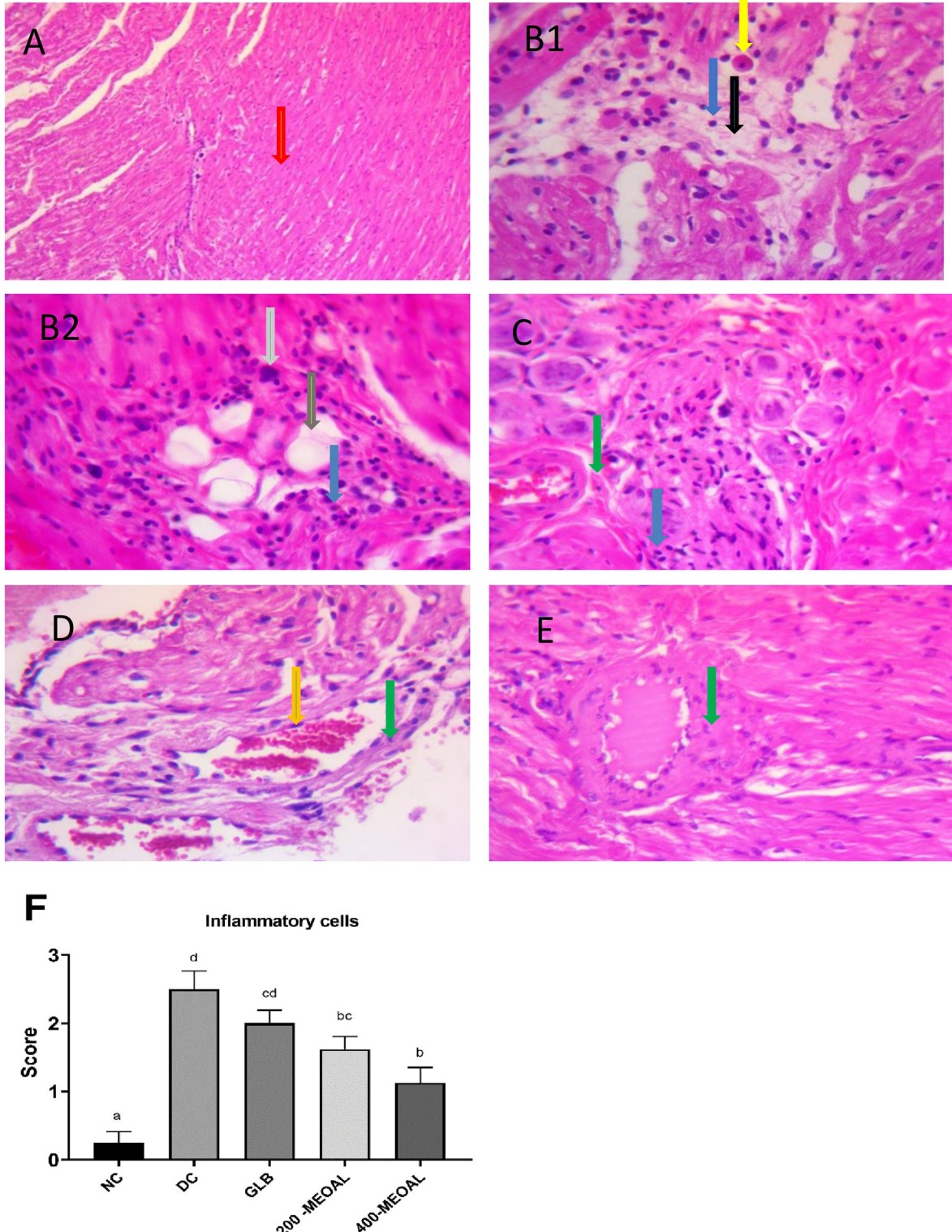

**Fig 4. Effect of GLB or MEOAL on histopathological changes of myocardial tissue in different groups.** Hematoxylin &Eosin staining sections under 40X magnification. https://doi.org/10.6084/m9.figshare.21828852.v1. Photomicrograph of myocardium from; (A) Normal control group showing healthy myocardial structure with normal myocyte (red arrow); (B1&B2) Diabetic control group showing significantly elevated in inflammatory cells infiltration (blue arrow) score compared to NC (P<0.05, F). Marked edema (black arrow) appeared with pyknosis of myocyte (yellow arrow). Some of the hypertrophic myocytes with enlarged nucleus (grey arrow) and fatty

change (brown arrow) were identified; (C) Diabetic GLB (5mg/kg) treated group showing mildly preserved myocardial structure with inhibition of ICS compared to DC (P>0.05, F); (D) Diabetic MEOAL (200mg/kg) treated group showing moderately preserved myocardial structure with inhibition of ICS compared to DC (P<0.05, F); (E) Diabetic MEOAL (400mg/kg) treated group showing dramatically preserved myocardial structure with inhibition of ICS compared to DC (P<0.05, F). However, Fibroblast cell formation (green arrow) and vascular congestion (orange arrow) were indicated in all treated diabetic groups. (F) The bar chart shows the significant differences between the groups in the inflammatory cells score. The results are expressed as mean± SEM (n = 8). P-value was calculated using ANOVA with Post Hoc Tukey test. The same letters mean no significant difference, the different letters (a, b, c, d) mean a significant difference at P<0.05. Abbreviations: NC; normal control group. DC; diabetic control group. ICS; Inflammatory cells score. GLB; 5mg/kg glibenclamide treated group .200or 400- MEOAL; 200 or 400mg/kg *O. acanthium* extract treated groups, respectively.

200mg/kg (Fig 4D). Statistical analysis showed significant inhibition of the inflammatory cell score (P<0.05) in the myocardial of diabetic rats treated with MEOAL (200 or 400mg/kg) as compared to the diabetic control group exhibiting their efficacy more robust than the effect of 5mg/kg glibenclamide (Fig 4C–4F).

**4. Immunohistochemical examination.** *4.1. Anti-insulin immunohistochemical expression of pancreatic islet tissue.* The positive insulin staining signals were demonstrated as dark brown granules observed in the cytoplasm of β-cells located in the center of the pancreatic islet. Immunohistochemical examination of the pancreatic section showed that the percentage area of insulin content was significantly lower in the diabetic control groups than in the normal control group (p<0.05) (Fig 5A, 5B & 5F). There was no significant increase in the percentage area of insulin content in glibenclamide or MEOAL (200 mg/kg) diabetic-treated groups compared to diabetic control group (p>0.05) (Fig 5C, 5D & 5F). Whereas the diabetic groups treated with MEOAL (400mg/kg) resulted in a significant rise in insulin content compared to the diabetic control group (p<0.05) (Fig 5E & 5F).

*4.2. Immunohistochemical expression of MRC1 (CD206) in macrophages M2 phenotype in myocardial tissue.* The expression of MRC1 (CD206) was slightly increased in the myocardium of the Glibenclamide-treated group compared to diabetic control group (P>0.05, Fig 6C & 6F), while there was a significant increase in positive CD206 macrophages M2 cells in myocardial heart tissue section of MEOAL -treated groups at both doses 200mglkg and 400mg/kg (P<0.05, Fig 6D, 6E & 6F), indicating M2 macrophage activation contributing to cardiac repair after myocardial damage and inflammation (Fig 6).

## Molecular docking study

The MEOAL contains six compounds Catechin, Epicatechin, Gallic Acid, kaempferol, P-Coumaric Acid, and Syringic acid. The binding site affinities were done between extract compounds with antidiabetic receptors: human pancreatic a-amylase target site (PDB ID: 2QV4). The binding affinity was highest for Catechin (−7.3 kcal/mol) followed by Epicatechin (−7.2 kcal/mol), Kaempferol (−7.1 kcal/mol), Gallic Acid (−6.1 kcal/mol), and Syringic Acid, P-Coumaric Acid (−5.8 kcal/mol). The binding mode orientation of the Catechin shows the formation of hydrogen bonds between Catechin with Asp 353, Asp 317, Arg 346, Arg 267 and Thr 314. Hydrophobic contact residues were established between Catechin with human pancreatic a-amylase. Catechin was also found to hydrophobic interaction with Arg 346 (Fig 7).

Kaempferol showed the highest binding site energy interaction with Human glucokinase PDB ID: 1V4S (−7.5 kcal/mol), which was lower than that of the reference standard, Glibenclamide (−8.9 kcal/mol). Kaempferol showed an extensive network of hydrogen bond interactions with Ser 445, Ser 151, Thr 228, and Arg 85, as well as Pi- cation and Pi anion interaction with Arg 85, and Asp 409, respectively. Amide-Pi stacked interaction was established with Gly80 (Table 6) (Fig 8).

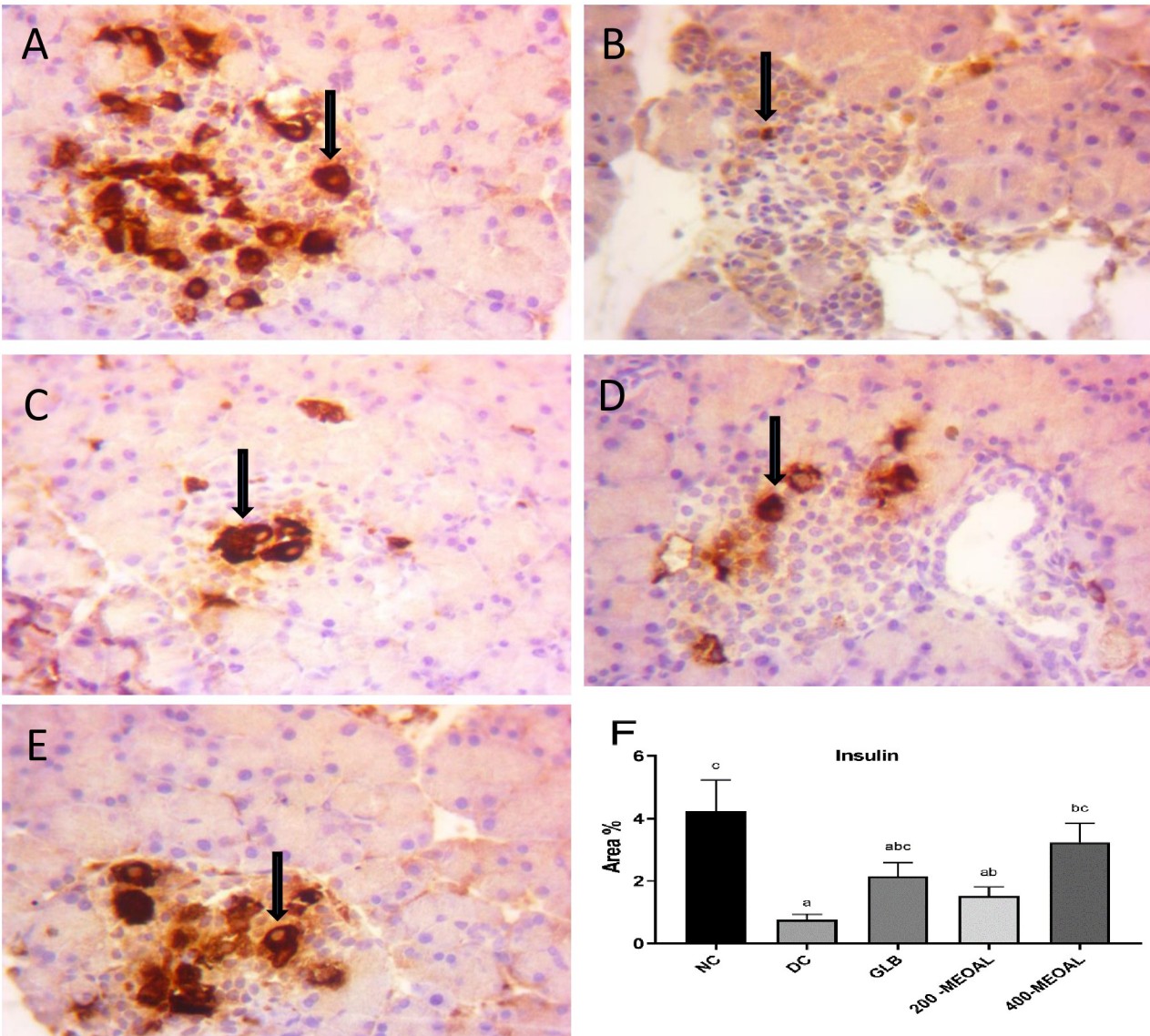

**Fig 5. Effect of GLB or MEOAL on insulin changes in the islets of Langerhans in different groups.** Anti-insulin Immunohistochemical staining (40 X). https://doi.org/10.6084/m9.figshare.21828864.v1. Pancreatic section from; (A) normal control group showing normal distribution of the insulin that occupies the center of the islet (arrow); (B) Diabetic control group showing marked reduction in the immunohistochemical expression of insulin in β-cells (arrow) compared to NC (P<0.05,F); (C) Diabetic GLB(5mg/kg) treated group showing an increase in insulin content of β-cells compared to DC (P>0.05,F); (D) Diabetic MEOAL (200mg/kg) treated group showing an increase of positive reactivity of insulin in the cytoplasm of β-cells (arrow) compared to DC(P>0.05,F); (E) Diabetic MEOAL (400mg/kg) treated group showing significant increase in the expression of insulin in β-cells (arrow) compared to DC (P<0.05,F). F: The bar chart represents anti-insulin immunopositive staining expressed as area%. The results are expressed as mean± ± SEM (n = 8). P-value was calculated using ANOVA with Post Hoc Tukey test. The same letters mean no significant difference, the different letters (a, b, c) mean a significant difference at P<0.05. Abbreviations: NC; normal control group. DC; diabetic control group. GLB; 5mg/ kg glibenclamide treated group .200or 400- MEOAL; 200 or 400mg/kg *O. acanthium* extract treated groups respectively.

The various binding free energy values of studied compounds showed in (Table 1) (-5.2 to -9.2 kcal/mol) and (-5.5 to -9.3 kcal/mol) suggests that they have a binding affinity to the Cyclooxygenase active site of COX-2 and Cyclooxygenase-1 COX-1, respectively. As a result of maximum binding score of -9.2 kcal/mol, Catechin was the most effective docked compound against (PDB:1CVU). Catechin interacts through a hydrogen bond with Asn 39, Asn 43, Gln

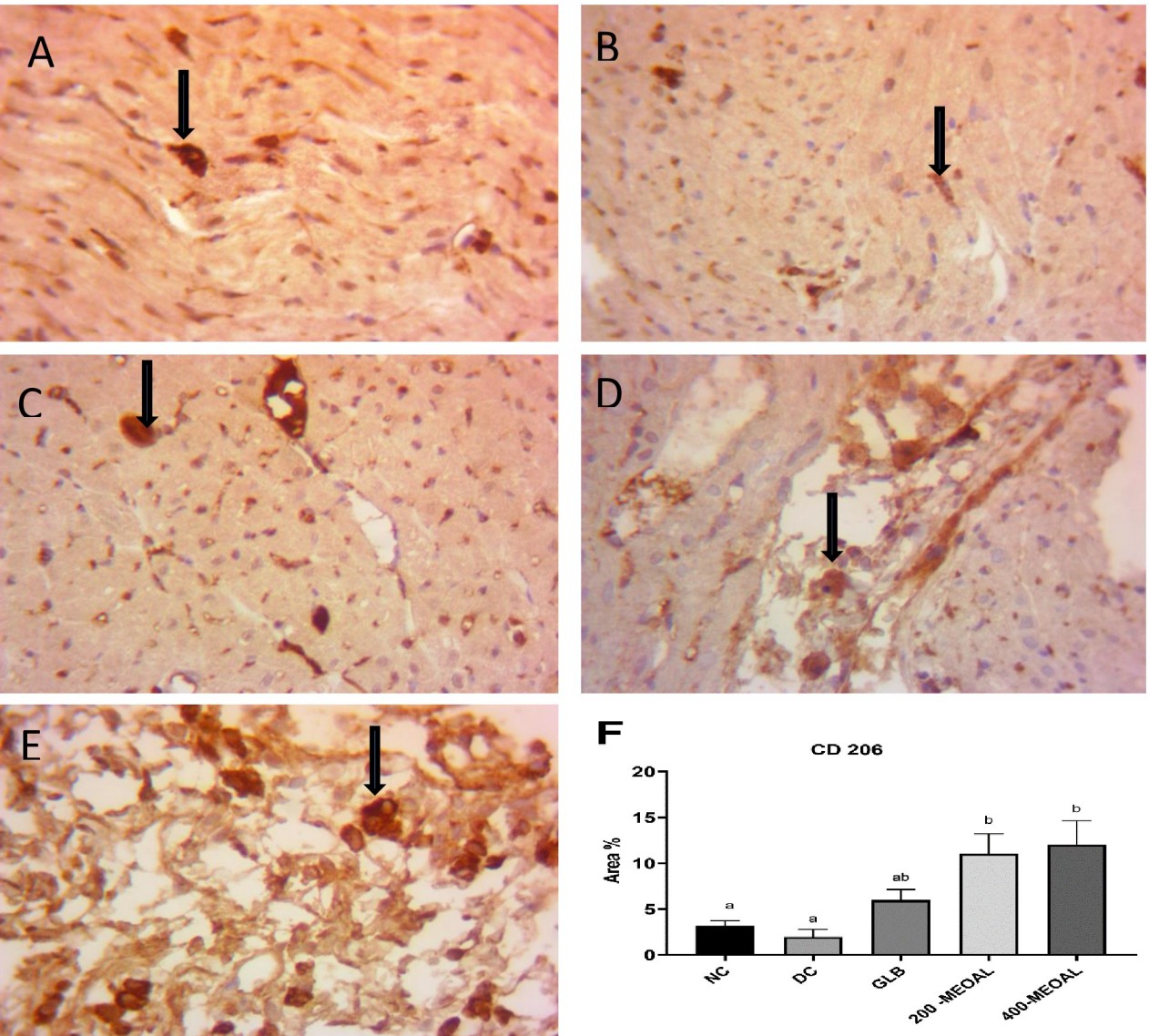

**Fig 6. Effect of GLB or MEOAL on CD206 cells positivity in the myocardium macrophage in different groups.** MRC1 Immunohistochemical staining (40X). https://doi.org/10.6084/m9.figshare.21828903.v1. Myocardium section from (A) Normal control group showing few expression of CD206 cells positivity (arrow); (B) Diabetic control group showing few expression of CD206 cells positivity (arrow); (C) Diabetic GLB(5mg/kg) treated group showing increase in CD206 cells positivity (arrow) compared to DC (P>0.05,F);(D) Diabetic MEOAL (200mg/kg) treated group showing significant increase in CD206 cells positivity (arrow) compared to DC (P<0.05,F); (E) Diabetic MEOAL (400mg/kg) treated group showing significant increase in CD206 cells positivity (arrow) compared to DC (P<0.05,F). F: The bar chart represents MRC1 immunopositive staining expressed as area%. The results are expressed as mean± SEM (n = 8). P-value was calculated using ANOVA with Post Hoc Tukey test. The same letters mean no significant difference, the different letters (a, b) mean a significant difference at P<0.05. Abbreviations: NC; normal control group. DC; diabetic control group. GLB; 5mg/kg glibenclamide treated group .200or 400- MEOAL; 200 or 400mg/kg *O. acanthium* extract treated groups respectively.

42, Gln 461, and Cys 47. The other amino acids residue in the binding pockets are Arg 469, Pro 153, and Cys 47 by pi-alkyl and Arg 44 by pi-hydrogen interaction (Fig 9).

Kaempferol, and Epicatechin showed favourable binding energy ranging from -8.7 kcal/mol to -7.6 Kcal/mol compared to the reference drug Glibenclamide with -7.7 Kcal/mol.

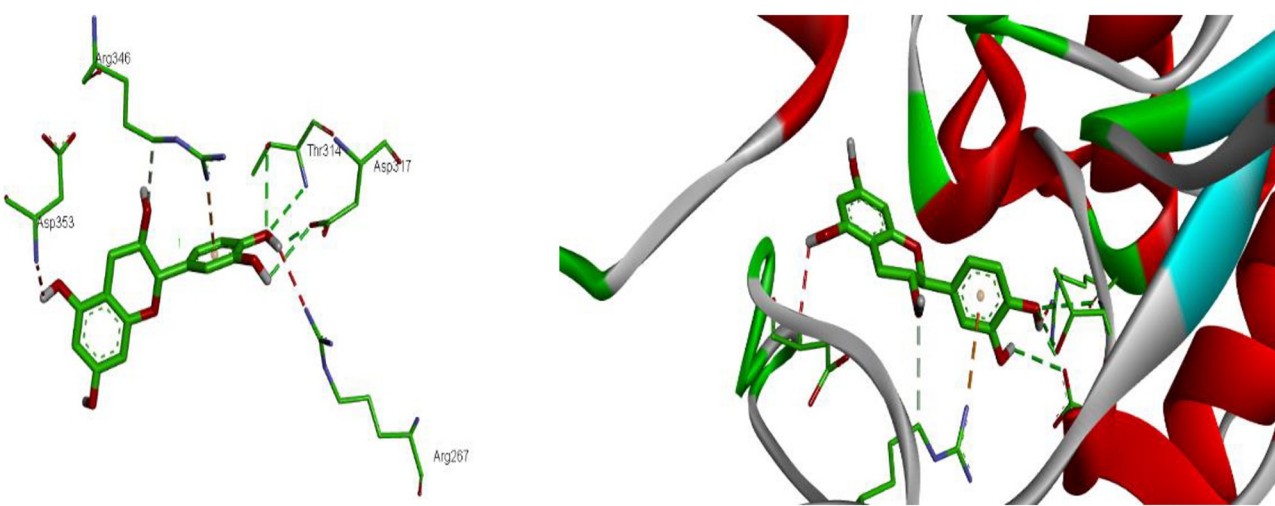

**Fig 7. 2D and 3D representation of the interaction of catechin with human pancreatic alpha-amylase (PDB ID: 2QV4).** https://doi.org/10.6084/m9.figshare.21828909.v1.

Kaempferol has shown the highest binding site of −9.3 kcal/mol against the receptors of COX-1(PDB ID: 3N8Y) (Table 7). Kaempferol binds to the active site of COX-1 through hydrogen bond with Cys 47, Cys36, Gly 45, Glu465, Gln461. The other amino acids residues present in the binding pocket of COX-1 are Ile 46, Leu 152, Pro 153 Ser420 in 3FCU and, ILE46, PRO153, LEU152. The differences of binding energy of Kaempferol (-9.3 kcal/mol), Catechin (-8.4 kcal/mol), and Epicatechin (-8.4 kcal/mol) suggests that they have the stronger binding affinity than the used standard drug Glibenclamide (-7.4 kcal/mol) (Fig 10).

## Discussion

In the present investigation, pancreatic degeneration and myocardium inflammation became risk factors in diabetic rats for eight weeks after STZ injection. The development in the pathogenesis of diabetic complications is commonly associated with a variety of contempt, including oxidative stress [32]. Thus, we estimated the hyperglycemic and toxic effect of STZ on rat heart and pancreas and studied the antidiabetic and antioxidant activities of MEOAL against oxidative stress and tissue damage through its phytochemical actions. Therefore, our focus is on the

**Table 6. In silico molecular docking results of GLB and bioactive phytochemicals interaction with antidiabetic receptors.**

| Compounds | Human pancreatic alpha-amylase (PDB ID: 2QV4) | Predicted Inhibition Constant p*K*i (µM) | Human glucokinase PDB ID: 1V4S | Predicted Inhibition Constant p*K*i (µM) |
|---|---|---|---|---|
| | Binding Energy (ΔG) (kcal/mol) | | Binding Energy (ΔG) (kcal/mol) | |
| Glibenclamide | **-7.7** | **6.1** | **-8.9** | **6.6** |
| Catechin | -7.3 | 5.9 | -7.4 | 6.0 |
| Kaempferol | -7.1 | 5.9 | -7.5 | 6.0 |
| Syringic Acid | -5.8 | 5.3 | -6.0 | 5.4 |
| P-Coumaric Acid | -5.8 | 5.3 | -6.0 | 5.4 |
| Epicatechin | -7.2 | 5.9 | -7.4 | 6.0 |
| Gallic Acid | -6.1 | 5.4 | -6.4 | 5.6 |

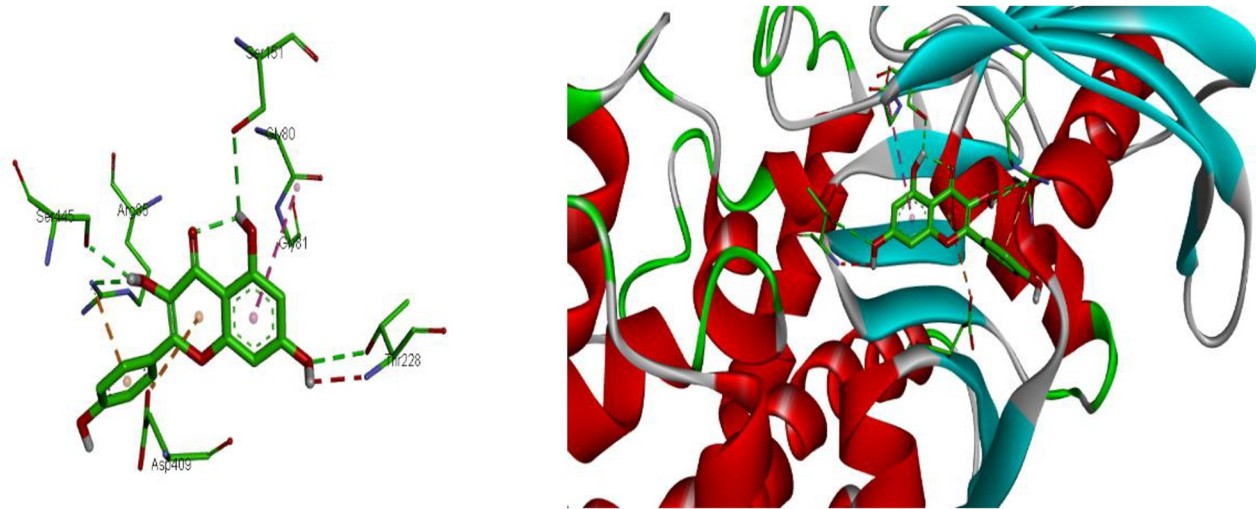

**Fig 8. 2D and 3D representation of the interaction of kaempferol with human glucokinase PDB ID: 1V4S.** https://doi.org/10.6084/m9.figshare.21828957.v1.

histopathological evaluation, which was confirmed by immunohistochemical analysis of the mentioned organs vulnerable to hyperglycemia.

Medicinal plants are widely used in the treatment of DM. The current study was the first investigation dealing with the hypoglycaemic effects of methanolic extract from leaves of *O. acanthium*, which enhance blood glucose levels after two, four, and eight weeks at both 200 and 400 mg/kg doses. This reduction may be attributed to the inhibitory effect of phenolic and flavonoid components (Table 1) on glucose absorption [32]. or through their capacity to induce insulin secretion from the remnant pancreatic β-cells [33], which share the same mechanism with the standard drug, glibenclamide. *O. acanthium* leaves are rich in glycoside forms of apigenin, quercetin, and luteolin, which play a crucial role in rescuing β-cells from

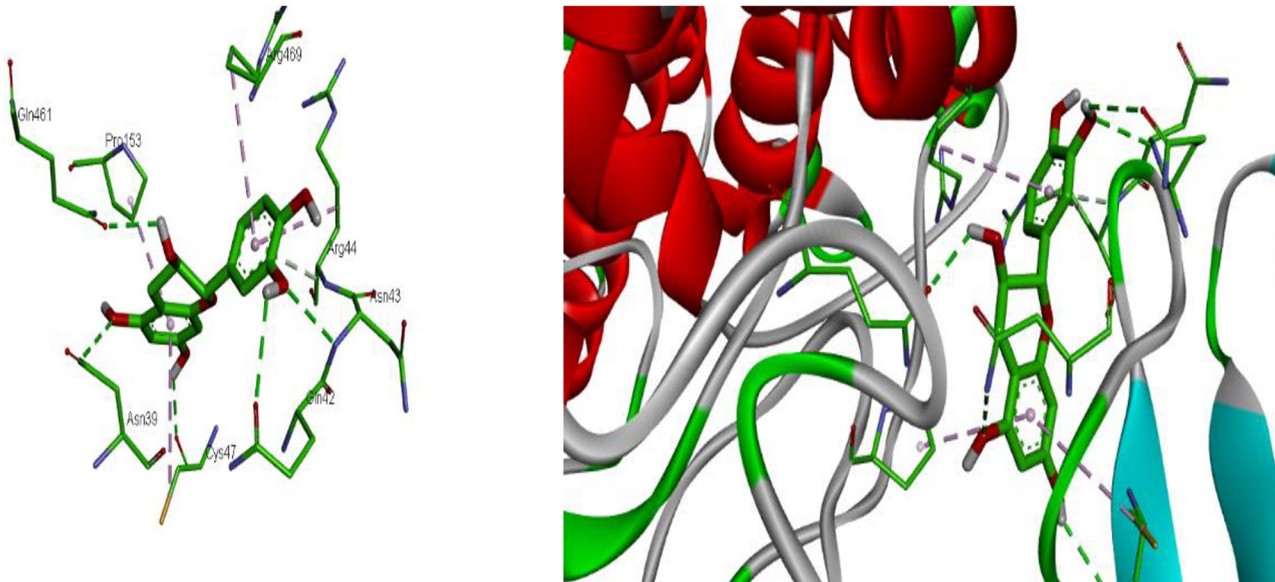

**Fig 9. 2D and 3D representations of the interaction of catechin with COX-2 (PDB ID: 1CVU).** https://doi.org/10.6084/m9.figshare.21828963.v1.

Table 7. **In silico molecular docking results of GLB and bioactive phytochemicals interaction with inflammation responsible receptors.**

| Compounds | COX-2 (PDB ID:1CVU) | Predicted Inhibition Constant pKi (µM) | COX-1(PDB ID: 3N8Y) | Predicted Inhibition Constant pKi (µM) |
|---|---|---|---|---|
| | Binding Energy (ΔG) (kcal/mol) | | Binding Energy (ΔG) (kcal/mol) | |
| glibenclamide | -7.7 | 6.1 | -7.4 | 5.9 |
| Catechin | -9.2 | 6.8 | -8.4 | 6.4 |
| Kaempferol | -8.7 | 6.6 | -9.3 | 6.8 |
| Syringic Acid | -5.2 | 5.0 | -6.0 | 5.4 |
| P-Coumaric Acid | -5.4 | 5.1 | -5.6 | 5.2 |
| Epicatechin | -7.6 | 6.0 | -8.4 | 6.4 |

destruction [34]. HPLC analysis confirmed the presence of kaempferol and p-coumaric acid in MEOAL reported for its antioxidant defenses against pancreatic β-cell damage [35, 36]. These mechanisms explain the possible control of glycemic levels in the blood, leading to attenuated body weight loss, which was monitored eight weeks after diabetic confirmation. The loss in body weight of diabetic rats is a common abnormality due to a metabolic imbalance of protein [37].

STZ is the most common diabetic inducer for its pancreatic β-cell cytotoxicity and diabetic activity through DNA alkylation-mediated necrosis, resulting in hyperglycemia. Pancreatic sections from diabetic rats possessed degeneration of β-cells, the necrotic appearance of islets, and abnormal architecture of most islet cells with low insulin immunoreactivity compared with normal islets. worth mentioning, treatment with MEOAL has almost regenerated damaged β-cells and normalized their insulin. These findings suggest that extract has several phenolic acids like gallic acid, syringic acid, and other bioactive molecules viz. epicatechin. Thus, these active compounds play a crucial role in the prevention of pathogenic processes related to pancreatic tissue through different action mechanisms. Latha and Daisy [38] reported that the biochemical mechanisms of gallic acid action may through improve the regeneration of the damaged β-cells of the pancreas in STZ-induced diabetic rats. in addition, gallic acid can improve RINm5f β-cell via its anti-apoptotic and insulin-secretagogue actions [39]. The

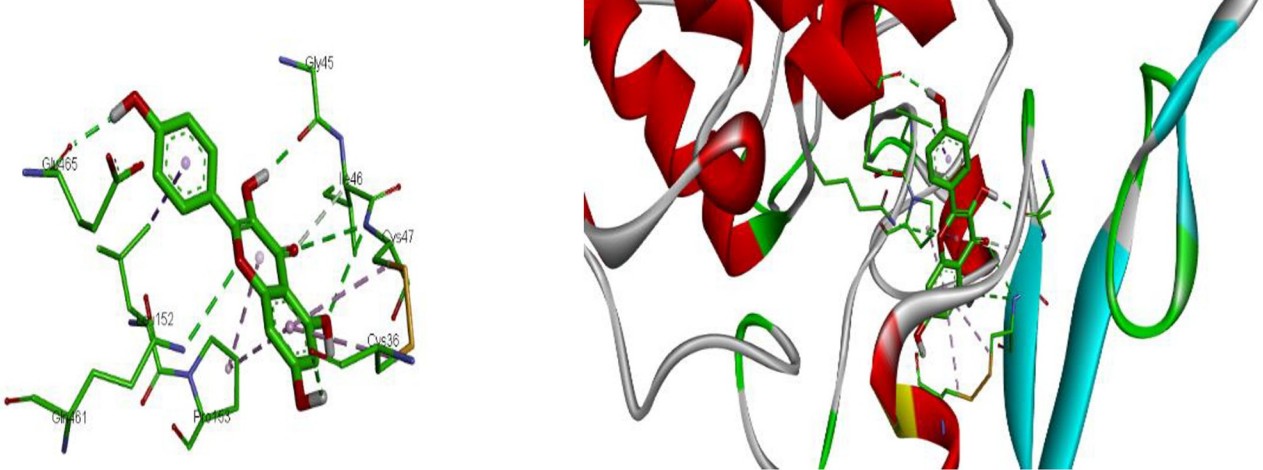

**Fig 10. 2D and 3D representations of the interaction of kaempferol with COX-1 (PDB ID: 3N8Y).** https://doi.org/10.6084/m9.figshare.21828969.v1.

presence of another phenolic compound in our extract as a syringic acid has an antihypergly-cemic effect through its potential to reduce pancreatic damage and stimulate β-cell regenera-tion. This was confirmed by the increase in the amount of immunoreactive insulin-secreting cells in pancreatic islets [40] To further evaluate the protective activities of MEOAL, Kim et al. examined the protective effects of flavonoids epicatechin on pancreatic islets and found this compound may suppress nitric oxide (NO) generation, which has been involved in β-cell dam-age during STZ induction [41]. Thus, the phenolic and flavonoid compositions participated together to recover the histological architecture of pancreatic tissue in diabetic rats, leading to increased insulin secretion, which profited blood glucose levels and body weight as shown in our results.

In addition, pancreatopathy hyperglycemia was also a major cause of DM complications and a risk factor for diabetic cardiomyopathy [42]. This is associated with myocardial inflam-mation, apoptosis, and oxidative stress [43]. The inflammatory process that occurs in DCM has a crucial role in histological and functional changes in a diabetic heart. ROS generation promotes the secretion of pro-inflammatory cytokines and the activation of macrophages [44]. In the present study, we demonstrated that MEOAL inhibited myocardial inflammation path-ologically and immunohistochemically. Furthermore, our results indicated that MEOAL has compounds with high antioxidative activities, suggesting it scavenges ROS [44] and subse-quently reduces inflammation and cardiac cell death. Previous studies show other mechanisms that explain the anti-inflammatory effect of flavonoid and phenolic compounds in medicinal plants. It has been demonstrated that flavonoid treatments reduce the release of anti-inflam-matory cytokines (TNF-α, IL6) and ameliorate myocardial injuries in diabetic rats [45]. They were also able to inhibit Nf-KB pathways in inducing cardiac inflammation [46]. Furthermore, the polyphenol component of gallic acid has been also reported to have anti-inflammatory properties by increasing plasma micro miR-24 and miR-126 levels [47]. Thus, we may specu-late that the anti-inflammatory potency of MEOAL at the dosage of 200 or 400 mg/kg related to its content of bioactive compounds results in a minimal inflammation score, which was more effective than the standard drug, glibenclamide. This finding confirmed more accuracy by the elevated reactivity of CD206 positive cells (market of anti-inflammatory M2 macro-phage) in the heart of diabetic rats treated with MEOAL compared to diabetic control rats. The M2 macrophages are often affiliated with tissue repair because they can antagonize the function of M1 macrophages that aggravate tissue damage [48]. It has been indicated that M1 macrophages (pro-inflammatory) were elevated in the heart of diabetic rats however the pres-ent study did not demonstrate any expression of pro-inflammatory markers, suggesting an essential role of inflammation in the development of diabetic cardiomyopathy [49]. In line with this notion, results observed in diabetic rats treated with 200 and 400 mg/kg of MEOAL suggest that this extract attenuated the inflammatory process and then cardiomyopathy com-plications. Probably via its acute phase once bioactive compounds of extract present anti-inflammatory action that participate in tissue repair.

Hence, the phytochemical and HPLC analyses exert that the methanol extract from *O. acanthium* leaves contains bioactive compounds, including gallic acid, catechin, epicatechin, syringic acid, kaempferol, and p-coumaric acid. The strong antioxidant properties of these compounds play a crucial role in the management of DM and its related disorders. Thereby, MEOAL administration reversed many diabetic complications and attenuated pancreatic and cardiac disorders.

Flavonoids have a wide range of antidiabetic actions where one flavonoid could target mul-tiple pathways [50]. Kaempferol has several antidiabetic effects, like improving AMP-activated cellular protein expression and activation, reducing cellular apoptosis by suppressing caspase 3 activities, and increasing the production and secretion of insulin from β-cells [51].

The protein-ligand complex is formed through the electrostatic interactions of the binding interface, including hydrogen bonds (both from side chains and backbones), salt bridges, and π-π stacking. Hydrogen bonding provides stability to protein molecules and selected protein-ligand interactions, thus being one of the most important for biological macromolecule interactions [30] pancreases and livers in a healthy condition secrete a highly localized enzyme which are essential for glucose metabolism, and regulation of insulin release by pancreatic β-cells. In the liver, glucokinase is involved in the process of glucose store as glycogen. The effect of MEOAL on the inhibition α-glucosidase and α-amylase activity displayed to be concentration dependent indicating the potential activity of extract's antidiabetic action [52]. Through *in silico* modelling six compounds in MEOAL are shown to bind to pancreatic alpha-amylase and glucokinase protein with Catechin, Kaempferol, and Epicatechin showing a highest binding affinities. The presence of flavonol in the extract might have a direct relationship with the reduction of blood glucose level. Hyperglycemia prompts reactive oxygen species (ROS) generation by activating several pathways which has a direct link with oxidative stress and inflammation [53, 54]. Catechin, Kaempferol, and Epicatechin were found to have highest glide scores with COX-1 and COX-2 indicating anti-inflammatory potential. phenolic hydroxyl groups that present in mentioned compounds are able to stabilize the free radicals [55]. A radical form of the antioxidant is produced, which is stabilized by charge delocalization caused by the interaction of the phenolic hydroxyl groups with the π-electrons of the benzene ring [56].

## Conclusions

We are the first to report that MEOAL has anti-diabetic activity in STZ- induced diabetic rats. The possible anti-diabetic mechanism of MEOAL was elucidated in that MEOAL attenuates pancreatic β-cell damage. This constructive effects of MEOAL could be attributed to the presence of bioactive compounds such as Catechin, Kaempferol, and Epicatechin which have highest binding site energy interaction with pancreatic alpha-amylase, and glucokinase receptors. Furthermore, the same compounds in extract have anti-inflammatory potential that inhibit COX-2, and COX-1 receptors. We suggest that the anti-inflammatory and antidiabetic efficacy of extract related to its anti-oxidant activity.

## Study limitation

Our study has some limitations, namely estimation of biochemical parameters such as serum insulin, glycated hemoglobin, and homeostatic model assessment for insulin resistance (HMOA-IR) which were not performed through experiential period due to the lack of special kits and equipment in our laboratory.

## Supporting information

**S1 Data.**
(RAR)

## Author Contributions

**Data curation:** Abdalmuhaimen Yusif Sharef, Hiwa Omer Ahmad.

**Formal analysis:** Abdalmuhaimen Yusif Sharef, Rafal Abdulrazaq Alrawi, Hiwa Omer Ahmad.

**Funding acquisition:** Abdalmuhaimen Yusif Sharef.

**Investigation:** Hiwa Omer Ahmad.

**Methodology:** Abdalmuhaimen Yusif Sharef, Rafal Abdulrazaq Alrawi, Hiwa Omer Ahmad.

**Project administration:** Abdalmuhaimen Yusif Sharef.

**Supervision:** Bushra Ahmed Hamdi, Rafal Abdulrazaq Alrawi.

**Validation:** Abdalmuhaimen Yusif Sharef, Rafal Abdulrazaq Alrawi, Hiwa Omer Ahmad.

**Visualization:** Abdalmuhaimen Yusif Sharef, Hiwa Omer Ahmad.

**Writing – original draft:** Abdalmuhaimen Yusif Sharef, Hiwa Omer Ahmad.

**Writing – review & editing:** Abdalmuhaimen Yusif Sharef, Hiwa Omer Ahmad.

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
