## [Decision Letter · Decision Letter 0]

17 Oct 2022

PONE-D-22-24149Onopordum acanthium extract attenuates pancreatic β-Cells and cardiac inflammation in streptozotocin-diabetic RatsPLOS ONE

Dear Dr. Bushra Ahmed Hamdi,

Thank you for submitting your manuscript to PLOS ONE. After careful consideration, we feel that it has merit but does not fully meet PLOS ONE’s publication criteria as it currently stands. Therefore, we invite you to submit a revised version of the manuscript that addresses the points raised during the review process.

We look forward to receiving your revised manuscript.

Kind regards,

Giribabu Nelli

Academic Editor

PLOS ONE

Journal Requirements:

Additional Editor Comments:

PONE-D-22-24149

"Onopordum acanthium extract attenuates pancreatic β-Cells and cardiac inflammation in streptozotocin-diabetic Rats"

Original Submission

Reviewer Recommendation Term: Major Revision

Rate Review: 0

Custom Review Question(s): Response

Comments to the Author

1. Is the manuscript technically sound, and do the data support the conclusions?

The manuscript must describe a technically sound piece of scientific research with data that supports the conclusions. Experiments must have been conducted rigorously, with appropriate controls, replication, and sample sizes. The conclusions must be drawn appropriately based on the data presented. Partly

2. Has the statistical analysis been performed appropriately and rigorously? Yes

3. Have the authors made all data underlying the findings in their manuscript fully available?

The PLOS Data policy requires authors to make all data underlying the findings described in their manuscript fully available without restriction, with rare exception (please refer to the Data Availability Statement in the manuscript PDF file). The data should be provided as part of the manuscript or its supporting information, or deposited to a public repository. For example, in addition to summary statistics, the data points behind means, medians and variance measures should be available. If there are restrictions on publicly sharing data—e.g. participant privacy or use of data from a third party—those must be specified. Yes

4. Is the manuscript presented in an intelligible fashion and written in standard English?

PLOS ONE does not copyedit accepted manuscripts, so the language in submitted articles must be clear, correct, and unambiguous. Any typographical or grammatical errors should be corrected at revision, so please note any specific errors here. No

5. Review Comments to the Author

Please use the space provided to explain your answers to the questions above. You may also include additional comments for the author, including concerns about dual publication, research ethics, or publication ethics. (Please upload your review as an attachment if it exceeds 20,000 characters) Thank you for inviting me to review this manuscript. I have the following comments:

1. Please, concise the abstract. It is too long.

2. The sentence lines 36-38 is an incomplete. Please, revise it.

3. Line 86: please, write β-cells in all manuscript sections. Not B-cells.

4. Line 193: Please, add ‘’Ethical approval’’ subtitle.

5. Line 215: replace the word ‘’divided’’ with the word ‘’allocated’’.

6. Line 232: in ‘’Experimental design’’, explain that 32 diabetic rats allocated into four groups other than the control non-diabetic (the fifth group).

7. Line 249: Please, replace the word ‘’measurement’’ with ‘’determination’’.

8. In Table 1, the authors used MRC1 Antibody as a human peptide, while your experiment was done on rats. Please, explain?????

9. In molecular docking, please, make it with protein of enzymes only, where you will determine if the ligands act as inhibitors or not.

10. Another comment regarding the molecular docking, the author made docking with protein of other species other than rat. Please, repeat it with rat proteins because your experiment was done on rats. But if you did not find the protein of rats, you can use the protein of human and make sequence alignment in https://www.uniprot.org/ to determine the degree of similarity between rat and human sequences of the same protein.

11. Please, write the pKi in Table 6 and 7.

6. PLOS authors have the option to publish the peer review history of their article (what does this mean?). If published, this will include your full peer review and any attached files.

Do you want your identity to be public for this peer review? For information about this choice, including consent withdrawal, please see our Privacy Policy.

No

Confidential to Editor

1. Do you have any potential or perceived competing interests that may influence your review? Please review our Competing Interests policy and declare any potential interests that you feel the Editor should be aware of when considering your review. If you have no competing interests, please write "I have no competing interests." Thank you for inviting me to review this manuscript. I have the following comments:

1. Please, concise the abstract. It is too long.

2. The sentence lines 36-38 is an incomplete. Please, revise it.

3. Line 86: please, write β-cells in all manuscript sections. Not B-cells.

4. Line 193: Please, add ‘’Ethical approval’’ subtitle.

5. Line 215: replace the word ‘’divided’’ with the word ‘’allocated’’.

6. Line 232: in ‘’Experimental design’’, explain that 32 diabetic rats allocated into four groups other than the control un-diabetic (the fifth group).

7. Line 249: Please, replace the word ‘’measurement’’ with ‘’determination’’.

8. In Table 1, the authors used MRC1 Antibody as a human peptide, while your experiment was done on rats. Please, explain?????

9. In molecular docking, please, make it with protein of enzymes only, where you will determine if the ligands act as inhibitors or not.

10. Another comment regarding the molecular docking, the author made docking with protein of other species other than rat. Please, repeat it with rat proteins because your experiment was done on rats. But if you did not find the protein of rats, you can use the protein of human and make sequence alignment in https://www.uniprot.org/ to determine the degree of similarity between rat and human sequences of the same protein.

11. Please, write the pKi in Table 6 and 7.

2. Did you receive any assistance in preparing this review (e.g. from a post-doc or graduate student)? If yes, please include their name below.

3. If accepted, do you think this submission should be highlighted on the PLOS ONE website? PLOS ONE does not evaluate manuscripts based on perceived significance or readership. We aim to provide tools for readers to filter and evaluate our publications. (optional)

Do you want to get recognition for this review on Publons?

If you opt in, your Publons profile will automatically be updated to show a verified record of this review in full compliance with the journal’s review policy. If you don’t have a Publons profile, you will be prompted to create a free account. Yes

 

PONE-D-22-24149

"Onopordum acanthium extract attenuates pancreatic β-Cells and cardiac inflammation in streptozotocin-diabetic Rats"

Original Submission

Reviewer Recommendation Term: Major Revision

Rate Review: 0

Custom Review Question(s): Response

Comments to the Author

1. Is the manuscript technically sound, and do the data support the conclusions?

The manuscript must describe a technically sound piece of scientific research with data that supports the conclusions. Experiments must have been conducted rigorously, with appropriate controls, replication, and sample sizes. The conclusions must be drawn appropriately based on the data presented. No

2. Has the statistical analysis been performed appropriately and rigorously? Yes

3. Have the authors made all data underlying the findings in their manuscript fully available?

The PLOS Data policy requires authors to make all data underlying the findings described in their manuscript fully available without restriction, with rare exception (please refer to the Data Availability Statement in the manuscript PDF file). The data should be provided as part of the manuscript or its supporting information, or deposited to a public repository. For example, in addition to summary statistics, the data points behind means, medians and variance measures should be available. If there are restrictions on publicly sharing data—e.g. participant privacy or use of data from a third party—those must be specified. Yes

4. Is the manuscript presented in an intelligible fashion and written in standard English?

PLOS ONE does not copyedit accepted manuscripts, so the language in submitted articles must be clear, correct, and unambiguous. Any typographical or grammatical errors should be corrected at revision, so please note any specific errors here. No

5. Review Comments to the Author

Please use the space provided to explain your answers to the questions above. You may also include additional comments for the author, including concerns about dual publication, research ethics, or publication ethics. (Please upload your review as an attachment if it exceeds 20,000 characters) Reviewer’s comments

I have read the manuscript entitled “Onopordum acanthium extract attenuates pancreatic β-Cells and cardiac inflammation in streptozotocin-diabetic Rats” several times and I have the following issues raised before the manuscript can be considered for possible publication

1. In the title the plant name is not the corrected way of writing. Authors should confirm and check the accepted using http://mpns.kew.org/mpns-portal/?_ga=1.111763972.1427522246.1459077346 or http://www.plantsoftheworldonline.org/ or http://www.worldfloraonline.org/.

- Also, in the title, streptozotocin-diabetic rats should be streptozotocin-induced diabetic rats

2. Abstract

-Did the author induced hypoglycemic associated inflammatory disorders

-the abbreviation of [OAE] does not represent methanolic extract of Onopordum acanthium (OAE).

- In the abstract line 28-30 should be rephrased and express correctly

-methods section not written properly. How many rats were used? How many groups did the authors used? Was there no standard drug group. If there was, how was the drug administered.

-line 36-38 not clear. Revise appropriately.

- Not enough biochemical assays was done to justify the study

-Under results insulin content, FBS and body weight alone is not enough to justify the study. More parameters need to be provided.

-line 42-43; STZ induced hyperglycemia associated inflammatory by the first sentence under background and objective shows it induces hypoglycemia associated inflammatory damage. Please clarify.

3. Introduction

-the accepted name of the plant should be written correctly.

- On the plant the anti-inflammatory activity has been reported before. What makes your work different from others.

-line 118 please revise

-line 122-123 should be moved above ……’To the best of our knowledge…..

Line 125-132 sentence does not make any sense. Authors should strive for clarity at all times. Revise

4. Methods

- The dosages of OAE 200 and 400 was not justified as well as the glibenclamide dose.

-DPPH assay to determine the antioxidant activity is not sufficient. Assays such as FRAP, Fe chelation, TAC, NO and OH radical scavenging should be carried out.

-Fasting blood glucose and body weight not enough.

-More parameters like serum insulin, glycated hemoglobin, HOMA-IR and HOMA-β should be carried out.

-in silico studies, the grid box for the docking should be provided.

-how were the ligands and proteins prepared. This should be separated and stated out clearly.

-Instead of the molecular docking, why didn’t the authors analyze for the induced-fit docking. Was the docking to the active site of the proteins or the authors did blind docking.

5. Discussion

- More recent studies can be used to improve the discussion especially the in silico section

6. Conclusion

-The analysis done is not sufficient enough to justify the conclusion made as well as the objectives of the study.

6. Conflict of interest

- The statement ‘The authors received no funding for this work’ is not appropriate here.

General comments

Authors should subject the manuscript to a native English speaker as most of the sentences are difficult to comprehend and also disjointed.

Thank you.

6. PLOS authors have the option to publish the peer review history of their article (what does this mean?). If published, this will include your full peer review and any attached files.

Do you want your identity to be public for this peer review? For information about this choice, including consent withdrawal, please see our Privacy Policy. No

Confidential to Editor

1. Do you have any potential or perceived competing interests that may influence your review? Please review our Competing Interests policy and declare any potential interests that you feel the Editor should be aware of when considering your review. If you have no competing interests, please write "I have no competing interests." No potential or perceived competing interests that may have influence my review.

2. Did you receive any assistance in preparing this review (e.g. from a post-doc or graduate student)? If yes, please include their name below. NO

3. If accepted, do you think this submission should be highlighted on the PLOS ONE website? PLOS ONE does not evaluate manuscripts based on perceived significance or readership. We aim to provide tools for readers to filter and evaluate our publications. (optional) Yes, on a more specific subject area page (e.g. Biochemistry, Atmospheric Science)

Do you want to get recognition for this review on Publons?

If you opt in, your Publons profile will automatically be updated to show a verified record of this review in full compliance with the journal’s review policy. If you don’t have a Publons profile, you will be prompted to create a free account.

Yes

 

PONE-D-22-24149

"Onopordum acanthium extract attenuates pancreatic β-Cells and cardiac inflammation in streptozotocin-diabetic Rats"

Original Submission

Reviewer Recommendation Term: Major Revision

Rate Review: 0

Custom Review Question(s): Response

Comments to the Author

1. Is the manuscript technically sound, and do the data support the conclusions?

The manuscript must describe a technically sound piece of scientific research with data that supports the conclusions. Experiments must have been conducted rigorously, with appropriate controls, replication, and sample sizes. The conclusions must be drawn appropriately based on the data presented. Partly

2. Has the statistical analysis been performed appropriately and rigorously? Yes

3. Have the authors made all data underlying the findings in their manuscript fully available?

The PLOS Data policy requires authors to make all data underlying the findings described in their manuscript fully available without restriction, with rare exception (please refer to the Data Availability Statement in the manuscript PDF file). The data should be provided as part of the manuscript or its supporting information, or deposited to a public repository. For example, in addition to summary statistics, the data points behind means, medians and variance measures should be available. If there are restrictions on publicly sharing data—e.g. participant privacy or use of data from a third party—those must be specified. Yes

4. Is the manuscript presented in an intelligible fashion and written in standard English?

PLOS ONE does not copyedit accepted manuscripts, so the language in submitted articles must be clear, correct, and unambiguous. Any typographical or grammatical errors should be corrected at revision, so please note any specific errors here. No

5. Review Comments to the Author

Please use the space provided to explain your answers to the questions above. You may also include additional comments for the author, including concerns about dual publication, research ethics, or publication ethics. (Please upload your review as an attachment if it exceeds 20,000 characters) Thank you for inviting me to review this manuscript. I have the following comments:

1. Please, concise the abstract. It is too long.

2. The sentence lines 36-38 is an incomplete. Please, revise it.

3. Line 86: please, write β-cells in all manuscript sections. Not B-cells.

4. Line 193: Please, add ‘’Ethical approval’’ subtitle.

5. Line 215: replace the word ‘’divided’’ with the word ‘’allocated’’.

6. Line 232: in ‘’Experimental design’’, explain that 32 diabetic rats allocated into four groups other than the control non-diabetic (the fifth group).

7. Line 249: Please, replace the word ‘’measurement’’ with ‘’determination’’.

8. In Table 1, the authors used MRC1 Antibody as a human peptide, while your experiment was done on rats. Please, explain?????

9. In molecular docking, please, make it with protein of enzymes only, where you will determine if the ligands act as inhibitors or not.

10. Another comment regarding the molecular docking, the author made docking with protein of other species other than rat. Please, repeat it with rat proteins because your experiment was done on rats. But if you did not find the protein of rats, you can use the protein of human and make sequence alignment in https://www.uniprot.org/ to determine the degree of similarity between rat and human sequences of the same protein.

11. Please, write the pKi in Table 6 and 7.

6. PLOS authors have the option to publish the peer review history of their article (what does this mean?). If published, this will include your full peer review and any attached files.

Do you want your identity to be public for this peer review? For information about this choice, including consent withdrawal, please see our Privacy Policy. No

Confidential to Editor

1. Do you have any potential or perceived competing interests that may influence your review? Please review our Competing Interests policy and declare any potential interests that you feel the Editor should be aware of when considering your review. If you have no competing interests, please write "I have no competing interests." Thank you for inviting me to review this manuscript. I have the following comments:

1. Please, concise the abstract. It is too long.

2. The sentence lines 36-38 is an incomplete. Please, revise it.

3. Line 86: please, write β-cells in all manuscript sections. Not B-cells.

4. Line 193: Please, add ‘’Ethical approval’’ subtitle.

5. Line 215: replace the word ‘’divided’’ with the word ‘’allocated’’.

6. Line 232: in ‘’Experimental design’’, explain that 32 diabetic rats allocated into four groups other than the control un-diabetic (the fifth group).

7. Line 249: Please, replace the word ‘’measurement’’ with ‘’determination’’.

8. In Table 1, the authors used MRC1 Antibody as a human peptide, while your experiment was done on rats. Please, explain?????

9. In molecular docking, please, make it with protein of enzymes only, where you will determine if the ligands act as inhibitors or not.

10. Another comment regarding the molecular docking, the author made docking with protein of other species other than rat. Please, repeat it with rat proteins because your experiment was done on rats. But if you did not find the protein of rats, you can use the protein of human and make sequence alignment in https://www.uniprot.org/ to determine the degree of similarity between rat and human sequences of the same protein.

11. Please, write the pKi in Table 6 and 7.

2. Did you receive any assistance in preparing this review (e.g. from a post-doc or graduate student)? If yes, please include their name below.

3. If accepted, do you think this submission should be highlighted on the PLOS ONE website? PLOS ONE does not evaluate manuscripts based on perceived significance or readership. We aim to provide tools for readers to filter and evaluate our publications. (optional)

Do you want to get recognition for this review on Publons?

If you opt in, your Publons profile will automatically be updated to show a verified record of this review in full compliance with the journal’s review policy. If you don’t have a Publons profile, you will be prompted to create a free account.

Yes

Reviewers' comments:

Reviewer's Responses to Questions

**Comments to the Author**

1. Is the manuscript technically sound, and do the data support the conclusions?

Reviewer #1: Yes

Reviewer #2: No

Reviewer #3: Partly

2. Has the statistical analysis been performed appropriately and rigorously? 

Reviewer #1: Yes

Reviewer #2: Yes

Reviewer #3: Yes

3. Have the authors made all data underlying the findings in their manuscript fully available?

Reviewer #1: Yes

Reviewer #2: Yes

Reviewer #3: Yes

4. Is the manuscript presented in an intelligible fashion and written in standard English?

Reviewer #1: Yes

Reviewer #2: No

Reviewer #3: No

5. Review Comments to the Author

Reviewer #1: The work entitled “Onopordum acanthium extract attenuates pancreatic β-Cells and cardiac inflammation in streptozotocin-diabetic Rats” is well designed and it contains different techniques including histology and immunohistochemistry, HPLC analyses, antioxidant activity and Molecular docking for testing of the anti-diabetic and cardio protective impact of extract from leaves of the O. acanthium. According to the authors’ results, the extract ameliorates negative effects of STZ. Finally, Authors discussed and proposed that rich bioactive compounds in the extract might have protective roles against to diabetic complications sourced from STZ. In my opinion, the manuscript can be accepted for publication but some minor remarks should be considered by authors before publication.

Minor comments:

Line 18-19: Delete the sententence "Streptozotocin (STZ) induces hypoglycaemia associated with inflammatory disorders." in Background and objective

Line 72: change the word “Diabetic” as “Diabetes”

Line 110: correct “( O.acanthium)” as “(O. acanthium)”

Line 136-137: Please indicate the name of the plant taxonomist who identified O. acanthium with a sentence.

Line 183: correct “per cent” as “percent”

Line 205: correct “Rattus norvegicus Domestica” as “Rattus norvegicus domestica” and as being italic.

Line 268: correct “deparaffinisation” as “deparaffinization”

Line 649: correct “Gallic“ as “gallic”

Line 687: correct “MiR-126” as “miR-126”

Reviewer #2: Reviewer’s comments

I have read the manuscript entitled “Onopordum acanthium extract attenuates pancreatic β-Cells and cardiac inflammation in streptozotocin-diabetic Rats” several times and I have the following issues raised before the manuscript can be considered for possible publication

1. In the title the plant name is not the corrected way of writing. Authors should confirm and check the accepted using http://mpns.kew.org/mpns-portal/?_ga=1.111763972.1427522246.1459077346 or http://www.plantsoftheworldonline.org/ or http://www.worldfloraonline.org/.

- Also, in the title, streptozotocin-diabetic rats should be streptozotocin-induced diabetic rats

2. Abstract

-Did the author induced hypoglycemic associated inflammatory disorders

-the abbreviation of [OAE] does not represent methanolic extract of Onopordum acanthium (OAE).

- In the abstract line 28-30 should be rephrased and express correctly

-methods section not written properly. How many rats were used? How many groups did the authors used? Was there no standard drug group. If there was, how was the drug administered.

-line 36-38 not clear. Revise appropriately.

- Not enough biochemical assays was done to justify the study

-Under results insulin content, FBS and body weight alone is not enough to justify the study. More parameters need to be provided.

-line 42-43; STZ induced hyperglycemia associated inflammatory by the first sentence under background and objective shows it induces hypoglycemia associated inflammatory damage. Please clarify.

3. Introduction

-the accepted name of the plant should be written correctly.

- On the plant the anti-inflammatory activity has been reported before. What makes your work different from others.

-line 118 please revise

-line 122-123 should be moved above ……’To the best of our knowledge…..

Line 125-132 sentence does not make any sense. Authors should strive for clarity at all times. Revise

4. Methods

- The dosages of OAE 200 and 400 was not justified as well as the glibenclamide dose.

-DPPH assay to determine the antioxidant activity is not sufficient. Assays such as FRAP, Fe chelation, TAC, NO and OH radical scavenging should be carried out.

-Fasting blood glucose and body weight not enough.

-More parameters like serum insulin, glycated hemoglobin, HOMA-IR and HOMA-β should be carried out.

-in silico studies, the grid box for the docking should be provided.

-how were the ligands and proteins prepared. This should be separated and stated out clearly.

-Instead of the molecular docking, why didn’t the authors analyze for the induced-fit docking. Was the docking to the active site of the proteins or the authors did blind docking.

5. Discussion

- More recent studies can be used to improve the discussion especially the in silico section

6. Conclusion

-The analysis done is not sufficient enough to justify the conclusion made as well as the objectives of the study.

6. Conflict of interest

- The statement ‘The authors received no funding for this work’ is not appropriate here.

General comments

Authors should subject the manuscript to a native English speaker as most of the sentences are difficult to comprehend and also disjointed.

Thank you.

Reviewer #3: Thank you for inviting me to review this manuscript. I have the following comments:

1. Please, concise the abstract. It is too long.

2. The sentence lines 36-38 is an incomplete. Please, revise it.

3. Line 86: please, write β-cells in all manuscript sections. Not B-cells.

4. Line 193: Please, add ‘’Ethical approval’’ subtitle.

5. Line 215: replace the word ‘’divided’’ with the word ‘’allocated’’.

6. Line 232: in ‘’Experimental design’’, explain that 32 diabetic rats allocated into four groups other than the control non-diabetic (the fifth group).

7. Line 249: Please, replace the word ‘’measurement’’ with ‘’determination’’.

8. In Table 1, the authors used MRC1 Antibody as a human peptide, while your experiment was done on rats. Please, explain?????

9. In molecular docking, please, make it with protein of enzymes only, where you will determine if the ligands act as inhibitors or not.

10. Another comment regarding the molecular docking, the author made docking with protein of other species other than rat. Please, repeat it with rat proteins because your experiment was done on rats. But if you did not find the protein of rats, you can use the protein of human and make sequence alignment in https://www.uniprot.org/ to determine the degree of similarity between rat and human sequences of the same protein.

11. Please, write the pKi in Table 6 and 7.

6. PLOS authors have the option to publish the peer review history of their article (what does this mean?). If published, this will include your full peer review and any attached files.

Reviewer #1: No

Reviewer #2: No

Reviewer #3: No

---

## [Author Response · Author response to Decision Letter 0]

10 Nov 2022

Dear Editors,

Thanks for this opportunity to publish our scientific research work in PLOS ONE, and thanks for your and the reviewers' valuable comments.

Responses to the reviewers' comments:

Reviewer #3

Thank you for inviting me to review this manuscript. I have the following comments:

Reviewer comment 1: Please, concise the abstract. It is too long.

Author response: Thanks for your scientific comment, we revised the abstract part based on your comment and highlighted in red colour.

 Reviewer comment 2: The sentence lines 36-38 is an incomplete. Please, revise it.

Author response: we changed all sentences and changed receptors.

Reviewer comment 3: Line 86: please, write β-cells in all manuscript sections. Not B-cells.

Author response: We corrected it accordingly and highlighted in red colour.

Reviewer comment 4: Line 193: Please, add ‘’Ethical approval’’ subtitle.

Author response: We added ethical approval subtitle based on your valuable comment and highlighted in red colour.

Reviewer comment 5: replace the word ‘’divided’’ with the word ‘’allocated’’.

Author response: We replaced based on your valuable comment and highlighted in red colour.

Reviewer comment 6: Line 232: in ‘’Experimental design’’, explain that 32 diabetic rats allocated into four groups other than the control non-diabetic (the fifth group).

Author response: We explained based on your suggestion and highlighted in red colour.

Reviewer comment 7: Line 249: Please, replace the word ‘’measurement’’ with ‘’determination’’.

Author response: We replaced accordingly and highlighted in red colour.

Reviewer comment 8: In Table 1, the authors used MRC1 Antibody as a human peptide, while your experiment was done on rats. Please, explain?????

Author response: Sorry for this mistake, we corrected and highlighted in red colour.

Reviewer comment 9: In molecular docking, please, make it with protein of enzymes only, where you will determine if the ligands act as inhibitors or not.

Author response: Protein molecules of Human pancreatic alpha-amylase (PDB ID: 2QV4), [1] Human glucokinase (PDB ID: 1V4S),[2] COX-2 (PDB ID: 1CVU), and COX-1 (PDB ID: 3N8Y)[3] were retrieved from the protein data bank (http://www.rcsb.org/pdb/) based on mentioned refrences.

Reviewer comment 10: Another comment regarding the molecular docking, the author made docking with protein of other species other than rat. Please, repeat it with rat proteins because your experiment was done on rats. But if you did not find the protein of rats, you can use the protein of human and make sequence alignment in https://www.uniprot.org/ to determine the degree of similarity between rat and human sequences of the same protein.

Author response: we have used proteins and used this website to compare receptors numbers of Amino acids in Human, rat, mouse, and sheep as requested https://www.uniprot.org/ to determine the degree of similarity 

 Antihyperglycemic [1] [2]

Human Rat

1 Human pancreatic alpha-amylase (PDB ID: 2QV4)

511 AA 508 AA

2 Human glucokinase (PDB ID: 1V4S) 

625 AA 627 AA

Note: We used proteins based on the above references, and there no much differences between Human and rat.

 Anti-inflammatory [3]

Human Rat mouse

1 Cyclooxygenase-2, COX-2 (PDB ID: 1CVU) Mus musculus (Mouse) (604 AA) PTGS2, COX2 (6014 AA) Ptgs2, Cox-2, Cox2 (604 AA) Ptgs2, Cox-2, Cox2, Pghs-b, Tis10 (604 AA)

2 COX-1 (PDB ID: 3N8Y) (600 AA)

 PTGS1, COX1 (599 AA) Ptgs1, Cox-1, Cox1 (602 AA) Ptgs1, Cox-1, Cox1 (6022 AA)

Note: We used receptors based on the above reference, and there no much differences between Human, sheep, Mouse, and rat.

Reviewer comment 11: Please, write the pKi in Table 6 and 7.

Author response: we have already calculated the pKi and written in Table 6 and 7 based on the reference that mentioned in manuscript. 

Reviewer #1

The work entitled “Onopordum acanthium extract attenuates pancreatic β-Cells and cardiac inflammation in streptozotocin-diabetic Rats” is well designed and it contains different techniques including histology and immunohistochemistry, HPLC analyses, antioxidant activity and Molecular docking for testing of the anti-diabetic and cardio protective impact of extract from leaves of the O. acanthium. According to the authors’ results, the extract ameliorates negative effects of STZ. Finally, Authors discussed and proposed that rich bioactive compounds in the extract might have protective roles against to diabetic complications sourced from STZ. In my opinion, the manuscript can be accepted for publication but some minor remarks should be considered by authors before publication.

Minor comments:

Reviewer comment 1: Line 18-19: Delete the sententence "Streptozotocin (STZ) induces hypoglycaemia associated with inflammatory disorders." in Background and objective

Author response: Thanks for your scientific comment; we deleted the sentence based on your valuable comment. 

Reviewer comment 2: Line 72: change the word “Diabetic” as “Diabetes”

Author response: We changed accordingly and highlighted in red colour.

Reviewer comment 3: Line 110: correct “( O.acanthium)” as “(O. acanthium)”

Author response: We corrected accordingly and highlighted in red colour.

Reviewer comment 4:. Line 136-137: Please indicate the name of the plant taxonomist who identified O. acanthium with a sentence.

Author response: We added a new sentence based on your valuable comment and highlighted in red colour.

Reviewer comment 5. Line 183: correct “per cent” as “percent”

Author response: We corrected accordingly and highlighted in red colour. 

Reviewer comment 6:.Line 205: correct “Rattus norvegicus Domestica” as “Rattus norvegicus domestica” and as being italic.

Author response: We corrected accordingly and highlighted in red colour.

Reviewer comment 7:.. Line 268: correct “deparaffinisation” as “deparaffinization”

Author response: We corrected accordingly and highlighted in red colour. 

Reviewer comment 8: Line 649: correct “Gallic“ as “gallic”

Author response: We corrected accordingly and highlighted in red colour. 

Reviewer comment 9:..Line 687: correct “MiR-126” as “miR-126”

Author response: We corrected accordingly and highlighted in red colour.

Reviewer #2

Reviewer’s comments

I have read the manuscript entitled “Onopordum acanthium extract attenuates pancreatic β-Cells and cardiac inflammation in streptozotocin-diabetic Rats” several times and I have the following issues raised before the manuscript can be considered for possible publication.

Reviewer comment 1 In the title the plant name is not the corrected way of writing. Authors should confirm and check the accepted using http://mpns.kew.org/mpns-portal/?_ga=1.111763972.1427522246.1459077346 or http://www.plantsoftheworldonline.org/ or http://www.worldfloraonline.org/.

Author response: Thanks for your scientific comment; we corrected the plant name accordingly and highlighted in red colour. 

Reviewer comment 2: Also, in the title, streptozotocin-diabetic rats should be streptozotocin-induced diabetic rats.

Author response: We changed it to streptozocin-induced diabetic rats, based on your valuable comment and highlighted in red colour.

2. Abstract

Reviewer comment 3:- Did the author induced hypoglycemic associated inflammatory disorders.

Author response: Sorry for this mistake, we deleted the sentence based on your valuable comment.

Reviewer comment 4:- the abbreviation of [OAE] does not represent methanolic extract of Onopordum acanthium (OAE).

Author response: We changed the abbreviation OAE into MEOAL( Methanol Extract of Onopordum acanthium Leaves) based on your suggestion and highlighted in red colour.

Reviewer comment 5- In the abstract line 28-30 should be rephrased and express correctly.

Author response: we have rephrased and used different receptors, thanks

Reviewer comment 6: Methods section not written properly. How many rats were used? How many groups did the authors used? Was there no standard drug group. If there was, how was the drug administered.

Author response: We revised methods section in the abstract part based on your comment and highlighted in red colour. 

Reviewer comment 7: line 36-38 not clear. Revise appropriately.

Author response: we have deleted it and used different design and receptors.

Reviewer comment 8: Not enough biochemical assays was done to justify the study.

Author response: Based on our topic, we focused on parameters such as insulin contain in pancreas , inflammatory cell score and MRC1 in cardiac tissue to explain the reduction of pancreatic damage and cardiac inflammation in treated diabetic rats. 

Reviewer comment 9: Under results insulin content, FBS and body weight alone is not enough to justify the study. More parameters need to be provided.

Author response: we estimated blood glucose and body weight during the eight weeks to confirm diabetic through this period. Our aim of the study was to evaluate the protective effect of extract against diabetic complications in pancreas and heart. 

Reviewer comment 10:-line 42-43; STZ induced hyperglycemia associated inflammatory by the first sentence under background and objective shows it induces hypoglycemia associated inflammatory damage. Please clarify.

Author response: Sorry for this mistake, we deleted the sentence based on your valuable comment.

3. Introduction

Reviewer comment 11: the accepted name of the plant should be written correctly.

Author response: we corrected the plant name accordingly and highlighted in red colour. 

Reviewer comment 12: On the plant the anti-inflammatory activity has been reported before. What makes your.

work different from others.

Author response: Our study exhibited anti-inflammatory effect of plant extract in Vivo as minimize the cardiac inflammation score associated diabetic complications for the first time. Furthermore, MRC1 staining conducted in this study to confirm tissue repair by phenolic compounds in extract.

Reviewer comment 13: line 118 please revise.

Author response: we revised the sentence based on your comment and highlighted in red colour.

Reviewer comment 14: line 122-123 should be moved above ……’To the best of our knowledge…..

Author response: we revised the sentence based on your comment and highlighted in red colour. 

Reviewer comment 15: Line 125-132 sentence does not make any sense. Authors should strive for clarity at all times. Revise

Author response: we have changed and highlighted all in red color. 

4. Methods

Reviewer comment 16: The dosages of OAE 200 and 400 were not justified as well as the glibenclamide dose.

Author response: We added references for extract and glibenclamide dose and highlighted in red colour.

Reviewer comment 17: DPPH assay to determine the antioxidant activity is not sufficient. Assays such as FRAP, Fe chelation, TAC, NO and OH radical scavenging should be carried out.

Author response: DPPH assay is available in our laboratory and we applied DPPH based on the following recent published works [4] [5], [6], [7]. 

Reviewer comment 18: Fasting blood glucose and body weight not enough.

Author response: our results showed diabetic complications histopathologically and immunohistochemically, and the docking study showed the binding site of phenolic compounds in extract on pancreatic and heart receptors (tissue) not in serum. Therefore fasting blood glucose and body weight parameters just to ensure diabetic continuous not diabetic complications. 

Reviewer comment 19: More parameters like serum insulin, glycated hemoglobin, HOMA-IR and HOMA-β should be carried out.

Author response: More parameters about diabetic such as serum insulin, glycated hemoglobin, HOMA-IR and HOMA-β should be carried out in this study, therefore we added this point to the limitation section. 

Reviewer comment 20: -in silico studies, the grid box for the docking should be provided.

Author response: we have provided the grid boxes for each receptor and highlighted in red colour as per requested. 

Reviewer comment 21: how were the ligands and proteins prepared. This should be separated and stated out clearly.

Author response: we have explained all steps of ligands and protein preparation in separate sections and highlighted in red color.

Reviewer comment 22: Instead of the molecular docking, why didn’t the authors analyze for the induced-fit docking. Was the docking to the active site of the proteins or the authors did blind docking.

Author response: we have done few restrictions on the research by changing the proteins based on the published papers as mentioned above and decreasing the receptors involved in inflammation to two receptors COX-1 and COX-2. 

5. Discussion

Reviewer comment 23: More recent studies can be used to improve the discussion especially the in-silico section

Author response: We have used recent studies to improve the discussion in silico section and highlighted in red color. 

6. Conclusion

Reviewer comment 24: The analysis done is not sufficient enough to justify the conclusion made as well as the objectives of the study.

Author response: we revised the whole conclusion section based on your comment and highlighted in red colour.

6. Conflict of interest

Reviewer comment 25: The statement ‘The authors received no funding for this work’ is not appropriate here.

Author response: we add (The authors declare that the research was conducted in the absence of any commercial or financial relationships that could be construed as a potential conflict of interest) under the title conflict of interest and highlighted on red colour.

General comments

Reviewer comment 26: Authors should subject the manuscript to a native English speaker as most of the sentences are difficult to comprehend and also disjointed.

Author response: Thanks for your valuable comment. We edited whole manuscript accordingly and highlighted in red. Please find the attached language editing certificate too.

All required rephrase and changes have been performed within manuscripts and highlighted in red color.

Thank you.

References 

1. Swilam, N., et al., Antidiabetic Activity and In Silico Molecular Docking of Polyphenols from Ammannia baccifera L. subsp. Aegyptiaca (Willd.) Koehne Waste: Structure Elucidation of Undescribed Acylated Flavonol Diglucoside. Plants, 2022. 11(3): p. 452.

2. Ab Rahman, N.S., et al., Molecular docking analysis and anti-hyperglycaemic activity of Synacinn™ in streptozotocin-induced rats. RSC advances, 2020. 10(57): p. 34581-34594.

3. Shahbazi, S., et al., Drug targets for cardiovascular-safe anti-inflammatory: In silico rational drug studies. PloS one, 2016. 11(6): p. e0156156.

4. Devequi-Nunes, D., et al., Chemical characterization and biological activity of six different extracts of propolis through conventional methods and supercritical extraction. PLoS One, 2018. 13(12): p. e0207676.

5. Juee, L.Y. and A.M. Naqishbandi, Calabash (Lagenaria siceraria) potency to ameliorate hyperglycemia and oxidative stress in diabetes. Journal of Functional Foods, 2020. 66: p. 103821.

6. Swaraz, A., et al., Phytochemical profiling of Blumea laciniata (Roxb.) DC. and its phytopharmaceutical potential against diabetic, obesity, and Alzheimer’s. Biomedicine & Pharmacotherapy, 2021. 141: p. 111859.

7. Nastasijević, B., et al., Inhibition of myeloperoxidase and antioxidative activity of Gentiana lutea extracts. Journal of pharmaceutical and biomedical analysis, 2012. 66: p. 191-196.

---

## [Decision Letter · Decision Letter 1]

2 Jan 2023

Onopordum acanthium L.extract attenuates pancreatic β-Cells and cardiac inflammation in streptozocin-induced diabetic rats

PONE-D-22-24149R1

Dear Dr. Bushra Ahmed Hamdi

We’re pleased to inform you that your manuscript has been judged scientifically suitable for publication and will be formally accepted for publication once it meets all outstanding technical requirements.

Kind regards,

Giribabu Nelli

Academic Editor

PLOS ONE

Additional Editor Comments (optional):

The manuscript can be accepted for publication.

Reviewers' comments:

Reviewer's Responses to Questions

**Comments to the Author**

1. If the authors have adequately addressed your comments raised in a previous round of review and you feel that this manuscript is now acceptable for publication, you may indicate that here to bypass the “Comments to the Author” section, enter your conflict of interest statement in the “Confidential to Editor” section, and submit your "Accept" recommendation.

Reviewer #1: All comments have been addressed

Reviewer #2: All comments have been addressed

2. Is the manuscript technically sound, and do the data support the conclusions?

Reviewer #1: Yes

Reviewer #2: Partly

3. Has the statistical analysis been performed appropriately and rigorously? 

Reviewer #1: Yes

Reviewer #2: Yes

4. Have the authors made all data underlying the findings in their manuscript fully available?

Reviewer #1: Yes

Reviewer #2: Yes

5. Is the manuscript presented in an intelligible fashion and written in standard English?

Reviewer #1: Yes

Reviewer #2: Yes

6. Review Comments to the Author

Reviewer #1: Authors have completed all recommended requirements by me. Thus, the manuscript can be accepted for publication in Plos One.

Reviewer #2: No further comments. All comments raised have been addressed. Other comments raised that cannot be addressed during revision has been duly addressed in the limitations of the study.

7. PLOS authors have the option to publish the peer review history of their article (what does this mean?). If published, this will include your full peer review and any attached files.

Reviewer #1: No

Reviewer #2: No

<quillbot-extension-portal></quillbot-extension-portal>

---

## [Editor Report · Acceptance letter]

16 Jan 2023

PONE-D-22-24149R1 

*Onopordum acanthium* L. extract attenuates pancreatic β-Cells and cardiac inflammation in streptozocin-induced diabetic rats 

Dear Dr. Hamdi:

I'm pleased to inform you that your manuscript has been deemed suitable for publication in PLOS ONE. Congratulations! Your manuscript is now with our production department. 

Kind regards, 

on behalf of

Dr. Giribabu Nelli 

Academic Editor

PLOS ONE